# Spectral Reach: Understanding Neural Scaling as Progress into the Spectral Tail

**Konstantin Nikolaou** [1] [2]   **Jonas Scheunemann** [1]   **Sven Krippendorf** [2]   **Samuel Tovey** [1]   **Christian Holm** [1]

## Abstract

Neural scaling laws describe predictable power-law relationships between model size, dataset size, compute, and performance. While these laws guide the development of modern foundation models, the mechanisms underpinning them remain poorly understood, in part due to the absence of scalable analysis tools. To close this gap, we introduce *spectral position*: a scalable measure of which eigenvalues of the empirical neural tangent kernel (eNTK) currently drive loss reduction. Applying this measure to scaling experiments, we find that spectral position decreases throughout training: learning shifts from dominant eigenmodes into the spectral tail. Larger models reach further into the tail than smaller models, revealing a size-dependent capacity we call *spectral reach*. This suggests why larger models achieve lower losses: they sustain learning on weak spectral signals inaccessible to smaller models. We further identify feature learning as a key enabler of spectral reach. It adaptively amplifies gradient magnitudes as learning advances, sustaining progress where frozen representations stall. This points to concrete interventions through architecture and optimizer design.

## 1. Introduction

The success of pre-training foundation models is largely attributed to the scaling paradigm: model quality improves predictably as training data, parameter count, and compute increase (Hestness et al., 2017; Kaplan et al., 2020; Hoffmann et al., 2022). This is quantitatively expressed in *neural scaling laws* which describe power-law relationships between scale variables (model size, dataset size, and com-

pute) and performance. In practice, these laws are central to pre-training large models: scaling curves are fit to computationally inexpensive runs and extrapolated to forecast the returns of additional compute. This guides choices such as model size, data allocation, and training duration (Kaplan et al., 2020; Hoffmann et al., 2022). Motivated by their practical importance, there has been significant interest in explaining the mechanisms behind scaling laws (Maloney et al., 2022; Bordelon et al., 2025; Bahri et al., 2024; Paquette et al., 2024; Sharma & Kaplan, 2022; Worschech & Rosenow, 2024; Cagnetta et al., 2025). However, many existing accounts rely on idealized assumptions or require kernel computations that are infeasible at modern scales, leaving open how scaling laws emerge in practical deep learning. To address this gap, we introduce a scalable diagnostic that reveals underlying spectral dynamics in large-scale training, and apply it to study the mechanisms of scaling in modern vision and language models.

We derive the loss-network-position (LNP) decomposition, which factors the instantaneous (linearized) loss change into three interpretable components: a network scale $\chi_{\text{net}}$, a loss scale $\chi_{\text{loss}}$, and a scale-free *spectral position* $\chi_{\text{pos}} \in [0, 1]$. Spectral position indicates which eigenvalues of the empirical neural tangent kernel (eNTK) currently drive loss reduction—close to 1 when dominant eigenmodes drive learning, close to 0 when learning focuses on the spectral tail. The factorization enables $\chi_{\text{pos}}$ to be evaluated from per-sample gradients without explicit kernel construction, permitting measurement alongside training with modest overhead. To validate the framework, we study controlled random-feature models, where $\chi_{\text{pos}}$ matches theory-predicted scaling relations.

Applying this diagnostic to scaling experiments reveals a consistent pattern (Figure 1): as training progresses, spectral position decreases, and larger, better-performing models reach *lower* values. A lower spectral position means learning from progressively smaller eigenvalues in the eNTK spectrum; larger models therefore have greater *spectral reach*. This suggests why they achieve lower losses: they sustain learning from weak spectral signals that smaller models cannot access.

A natural question is what enables spectral reach. Through linear-probing experiments, we find that feature learning

[1]Institute for Computational Physics, University of Stuttgart, Stuttgart, Germany [2]Department of Applied Mathematics and Theoretical Physics and Department of Physics, University of Cambridge, Cambridge, UK. Correspondence to: Konstantin Nikolaou <konsti.nikolaou@gmail.com>.

*Proceedings of the $43^{rd}$ International Conference on Machine Learning*, Seoul, South Korea. PMLR 306, 2026. Copyright 2026 by the author(s).

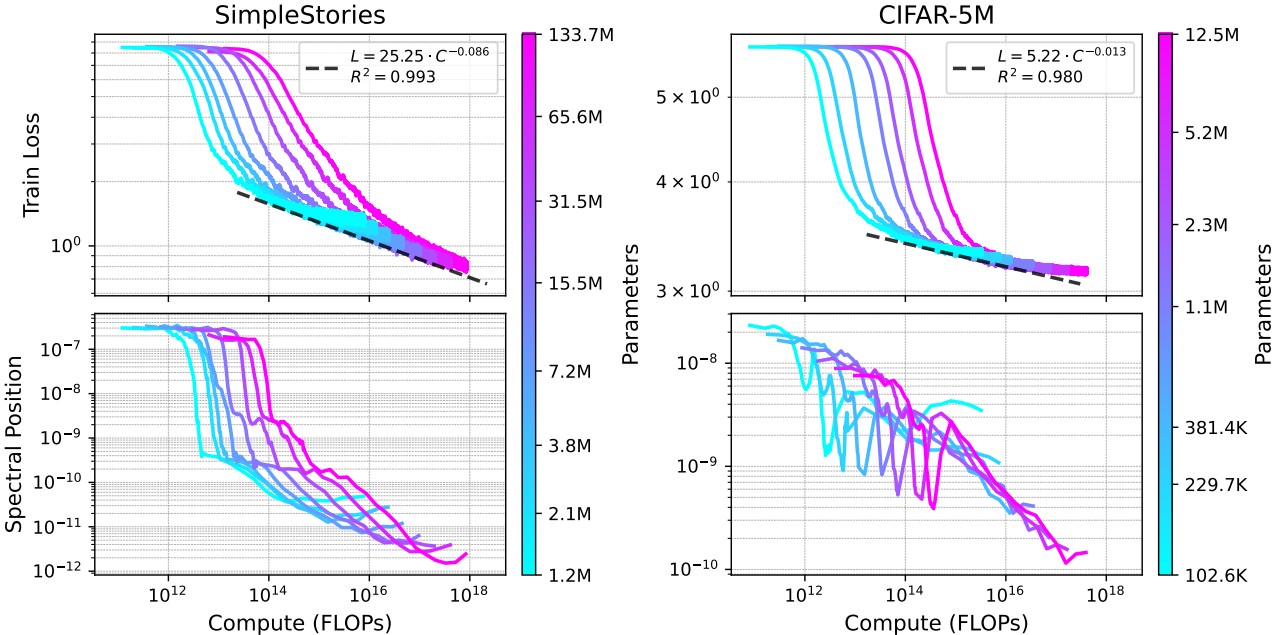

*Figure 1.* Scaling laws and spectral position for Llama 2 language models on SimpleStories (left) and next-pixel prediction on CIFAR-5M (right). Following Hoffmann et al. (2022), the compute is approximated as FLOPs = $6ND$, where $N$ is the number of model parameters and $D$ is the number of processed tokens/images. **(top)** Scaling laws emerge as power-law relations between model size and train loss. The dashed lines indicate power-law fits with exponents and $R^2$ values. We provide test losses in Appendix D.1. **(bottom)** Spectral position $\chi_{\text{pos}}$ decreases throughout training and with model size. Larger models achieve lower values.

plays a key role in spectral reach, and provide a mechanistic explanation: as spectral position decreases, feature-learning models adaptively amplify gradient magnitudes to sustain learning progress—a compensation mechanism unavailable to models with frozen representations. This opens practical avenues for enhancing spectral reach through targeted interventions, which we discuss in the Outlook.

## 2. Related Work

**Theoretical explanations of scaling (and the validation gap).** A major line of work proposes explanations by analyzing simplified but tractable surrogates and connecting scaling exponents to spectral structure in data and kernels (Maloney et al., 2022; Sharma & Kaplan, 2022; Lin et al., 2024). *Random feature models (RFMs)* are a prominent example and have become a testbed for deriving scaling behavior from first principles (Maloney et al., 2022; Bordelon et al., 2025; Bahri et al., 2024; Paquette et al., 2024). In RFMs, a feature map is drawn at random and kept fixed, and only a linear readout is trained. This reduction to linear regression makes training and generalization analytically tractable, with closed-form expressions for many relevant quantities (Yehudai & Shamir, 2019). RFMs reproduce power-law scaling curves, when the kernel is assumed to exhibit a power-law spectral structure (Maloney et al., 2022; Bordelon et al., 2025). At the same time, because represen-

tations are random and frozen, RFMs cannot capture the feature learning and hierarchical abstraction formation central to deep networks (Yehudai & Shamir, 2019; Cagnetta et al., 2025). Empirically, scaling curves in deep networks resemble those predicted by kernel surrogates, however, including feature learning and the nature of the power-law into the picture remains an open question (Bahri et al., 2024; Cagnetta et al., 2025).

Bordelon et al. (2020) studied dataset size scaling in kernel models, demonstrating that increasing data enables learning from modes with smaller eigenvalues in the kernel spectrum. Bahri et al. (2024) extended this idea by arguing that finite model capacity similarly induces an effective cutoff in the small-eigenvalue tail of the empirical kernel spectrum: Modes beyond this cutoff are effectively inaccessible, and the remaining (unlearned) tail governs the residual error. Thus, scaling improves performance by pushing this cutoff deeper, enabling the model to leverage progressively weaker spectral modes. While this reasoning was proposed for RFMs, it is unclear how feature learning interacts with this picture and whether this mechanism even applies to practical deep learning scenarios.

**NTK and scalable measurements.** The neural tangent kernel (NTK) offers a principled lens on learning dynamics: under suitable initialization, the infinite-width limit yields a

linearized training dynamics around the initial parameters, enabling kernel-based analysis (Jacot et al., 2018; Lee et al., 2020). In this ("lazy training") regime, the NTK is effectively constant (Chizat et al., 2019); by contrast, at finite width the NTK can evolve, reflecting representation learning and feature drift (Fort et al., 2020; Yang & Hu, 2021). To highlight the contrast between these regimes, we refer to the latter as the empirical NTK (eNTK).

However, direct computations of the eNTK remain limited to relatively small datasets and models. Even with efficient estimators (Novak et al., 2022) and sketching techniques (Han et al., 2021; Zandieh et al., 2021), the computational and memory costs are typically infeasible in the regimes where scaling laws are measured. Our work targets this measurement bottleneck. The loss-network-position (LNP) decomposition can be computed from per-sample gradients (Yousefpour et al., 2022) and enables tracking eNTK quantities at scales where explicit kernel computations are impractical.

**Empirical diagnostics of learning dynamics.** The question of which eigenmodes currently shape learning has been partially addressed in studies on kernel alignment, including kernel-target alignment (Cristianini et al., 2001) and centered alignment between kernel matrices for kernel learning (Cortes et al., 2012). Several empirical works report increasing eNTK alignment during training (Baratin et al., 2021; Atanasov et al., 2021; Shan & Bordelon, 2022; Ortiz-Jimenez et al., 2021). These results typically refer to kernel alignment with respect to *fixed* task-relevant directions, answering *has the model adapted to the task?*. In contrast, $\chi_{\text{pos}}$ (Section 3) answers a different question: *where in the eNTK spectrum is learning currently focused?* It weights eigenvalues by the projection of the *current* loss gradient.

Another related line of work studies the *spectral bias*—the empirical observation that neural networks tend to learn low-frequency functions before high-frequency ones (Rahaman et al., 2019; Canatar et al., 2021). This phenomenon has been linked to learning dynamics described by the NTK (Cao et al., 2020), where large-eigenvalue modes (corresponding to low-frequency functions) are learned more rapidly than small-eigenvalue modes.

Our work is also related to broader empirical diagnostics of large-scale optimization such as gradient noise scale (McCandlish et al., 2018; Cohen et al., 2020), curvature/sharpness proxies (Fort et al., 2020), and complexity measures like effective rank (Baratin et al., 2021; Nikolaou et al., 2026a), matrix entropy and measures of spectral magnitude (Tovey et al., 2023; 2025; Krippendorf & Spannowsky, 2022). These tools quantify aspects of optimization and generalization but do not directly measure *which spectral modes* remain learnable and are being utilized over

time.

Taken together, these threads highlight a recurring limitation: many mechanistic scaling hypotheses are naturally expressed in spectral or kernel terms, but the corresponding measurements are difficult to obtain at the scales where scaling laws are observed. Our work addresses this gap with a scalable diagnostic and uses it to study spectral learning dynamics in practical scaling experiments.

## 3. Cheap Measures through LNP-Decomposition

### 3.1. Derivation

In this section, we introduce a decomposition of the loss dynamics into three components: the loss-network-position (LNP) decomposition. It enables efficient computation of the spectral position, which otherwise requires expensive eNTK calculations.

For simplicity, we present the derivation for full-batch gradient descent. While stochastic mini-batch training introduces additional complexity, the core concepts extend naturally, as discussed in the "In Practice" paragraph below. The full derivation is provided in Appendix A.1.

Let $\mathbf{X} = [\mathbf{x}_1, ..., \mathbf{x}_D]^T$ denote the matrix of $D$ training inputs $\mathbf{x}_i$ and $\mathbf{Y} = [y_1, ..., y_D]^T$ be the corresponding target outputs, which can be either supervised labels or self-supervised targets. Let the neural network be represented by a function $f(\mathbf{x}; \boldsymbol{\theta})$ with parameters $\boldsymbol{\theta}$.

The training process involves minimizing a loss function $\mathcal{L}(f(\mathbf{X}; \boldsymbol{\theta}), \mathbf{Y}) := \mathcal{L}(\mathbf{X}, \boldsymbol{\theta})$, over the dataset.[1] During training, the parameters $\boldsymbol{\theta}$ are updated iteratively using gradient descent.

$$\Delta\boldsymbol{\theta}_t := \boldsymbol{\theta}_{t+1} - \boldsymbol{\theta}_t = -\eta\nabla_{\boldsymbol{\theta}}\mathcal{L}(\mathbf{X}, \boldsymbol{\theta}_t), \qquad (1)$$

where $\eta$ is the learning rate. Over the course of training, the loss decreases from its initial value $\mathcal{L}(\mathbf{X}, \boldsymbol{\theta}_0)$ to $\mathcal{L}(\mathbf{X}, \boldsymbol{\theta}_t)$ at training step $t$. The change of the loss to a parameter update is given by

$$\Delta\mathcal{L}(\mathbf{X}, \boldsymbol{\theta}_t) := \mathcal{L}(\mathbf{X}, \boldsymbol{\theta}_{t+1}) - \mathcal{L}(\mathbf{X}, \boldsymbol{\theta}_t). \qquad (2)$$

To disentangle the factors driving the loss decrease, we use a first-order Taylor expansion to distinguish linear and non-linear contributions to the loss change at each training step $t$. For sufficiently small learning rates $\eta$, the parameter updates

---

[1] For simplicity, we omit the explicit notation of targets, as each input has a corresponding pair.

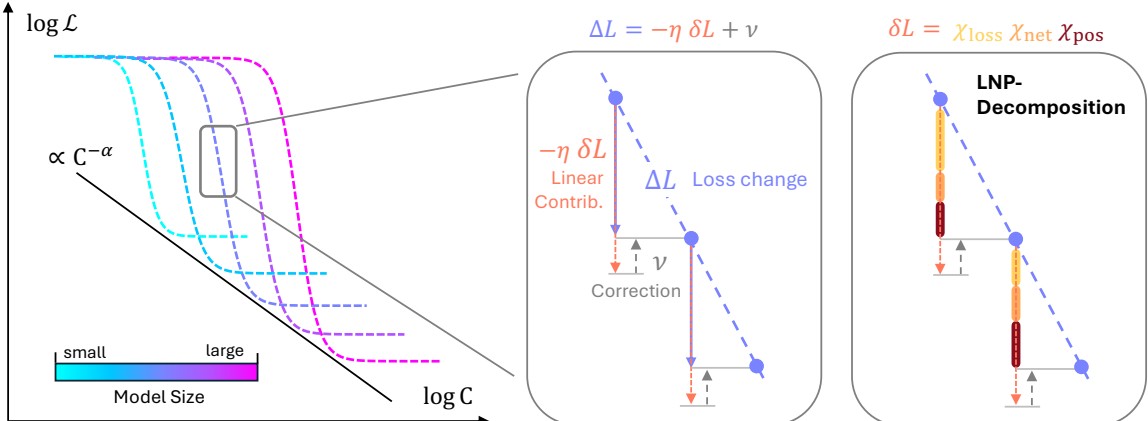

*Figure 2.* Intuition behind Loss-Network-Position (LNP) decomposition for neural scaling laws. **(left)** Schematic of training curves of different model sizes—larger models or more data lead to lower final losses. **(middle)** Decomposition of the loss evolution into linear contribution and corrections. The linear component captures the immediate effect of parameter updates on the loss, while corrections account for stochastic and non-linear effects. **(right)** Further decomposition of the linearized loss into LNP-components capturing contributions from loss function, the network architecture, and spectral position.

$\Delta\boldsymbol{\theta}_t$ are small, making this expansion accurate:

$$\Delta\mathcal{L}(\mathbf{X},\boldsymbol{\theta}_t) = \underbrace{\nabla_{\boldsymbol{\theta}}^T\mathcal{L}(\mathbf{X},\boldsymbol{\theta}_t)\,\Delta\boldsymbol{\theta}_t}_{\text{(linearized change)}} + \underbrace{O\Big(\|\Delta\boldsymbol{\theta}_t\|^2\Big)}_{:=\nu_t\ \text{(non-linear correction)}}$$

$$= -\eta\,\underbrace{\|\nabla_{\boldsymbol{\theta}}\mathcal{L}(\mathbf{X},\boldsymbol{\theta}_t)\|^2}_{=:\delta\mathcal{L}_t} + \nu_t \tag{3}$$

$$= -\eta\,\delta\mathcal{L}_t + \nu_t$$

We used the gradient descent update rule to express the linear component $\delta\mathcal{L}_t$ independently of the learning rate $\eta$, and a non-linear correction term $\nu_t$. An illustration of this decomposition is provided in Figure 2 (middle).

We decompose the linearized loss $\delta\mathcal{L}$ into three components representing distinct scale contributions: the network architecture, the loss function, and their interaction. This decomposition takes the form

$$\delta\mathcal{L} = (\nabla_{\boldsymbol{\theta}}\mathcal{L})^T(\nabla_{\boldsymbol{\theta}}\mathcal{L}) \tag{4}$$

$$= (\nabla_f\mathcal{L})^T\underbrace{\nabla_{\boldsymbol{\theta}}f(\nabla_{\boldsymbol{\theta}}f)^T}_{=:\,\Theta=\sum_k\lambda_k q_k q_k^T}\nabla_f\mathcal{L} \tag{5}$$

$$= \sum_k (\nabla_f^T\mathcal{L}\,q_k)^2\,\lambda_k \tag{6}$$

$$= \underbrace{\|\nabla_f\mathcal{L}\|_2^2}_{=:\,\chi_{\text{loss}}}\underbrace{\|\nabla_{\boldsymbol{\theta}}f\|_F^2}_{=:\,\chi_{\text{net}}}\underbrace{\sum_k\frac{(\nabla_f^T\mathcal{L}\,q_k)^2}{\|\nabla_f\mathcal{L}\|_2^2}\frac{\lambda_k}{\|\nabla_{\boldsymbol{\theta}}f\|_F^2}}_{=:\,\chi_{\text{pos}}} \tag{7}$$

$$= \chi_{\text{loss}}\cdot\chi_{\text{net}}\cdot\chi_{\text{pos}}, \tag{8}$$

where we have used $\nabla_{\boldsymbol{\theta}}\mathcal{L} = (\nabla_{\boldsymbol{\theta}}f)^T\nabla_f\mathcal{L}$ and defined the empirical NTK as $\Theta = \nabla_{\boldsymbol{\theta}}f(\nabla_{\boldsymbol{\theta}}f)^T$ with eigenmodes $q_k$ and eigenvalues $\lambda_k$. We denote the Frobenius norm by $\|\cdot\|_F$

and the Euclidean norm by $\|\cdot\|_2$. The key steps involve projecting the loss evolution onto the eigenmodes of the eNTK and factoring out scale contributions from the network and the loss function.

**The Magnitudes $\chi_{\text{net}}$ and $\chi_{\text{loss}}$** capture the individual scales of the network and loss function: $\chi_{\text{net}}$ quantifies the overall sensitivity of the network outputs to parameter changes, reflecting how responsive the model is to parameter updates. It is equivalent to the trace of the eNTK matrix $\chi_{\text{net}} = \text{Tr}(\Theta)$. $\chi_{\text{loss}}$ captures the magnitude of the loss gradient with respect to the network outputs, reflecting how sensitive the loss is to changes in predictions. This term only depends on the specific loss function used (e.g., MSE, cross-entropy) and the current prediction errors.

**The Spectral Position $\chi_{\text{pos}}$** is a scale-free quantity with range $\chi_{\text{pos}} \in [0, 1]$ that indicates which eigenvalues of the eNTK currently drive loss reduction. To understand its structure, we can rewrite it as

$$\chi_{\text{pos}} = \sum_k\underbrace{\frac{(\nabla_f^T\mathcal{L}\,q_k)^2}{\|\nabla_f\mathcal{L}\|_2^2}}_{=:\,\gamma_k^2}\underbrace{\frac{\lambda_k}{\|\nabla_{\boldsymbol{\theta}}f\|_F^2}}_{=:\,p_k} = \sum_k\gamma_k^2\,p_k, \quad (9)$$

where $\gamma_k$ are the *projection coefficients* quantifying how much of the loss gradient lies along the $k$-th eNTK eigenmode, and $p_k$ are the *normalized eigenvalues* representing the relative strength of each eNTK mode. Note that $\sum_k\gamma_k^2 = \sum_k p_k = 1$, so $\gamma_k^2$ re-weights the eNTK spectrum according to where the loss gradient projects, and $\chi_{\text{pos}}$ extracts the expected (normalized) eigenvalue that drives loss reduction.

The range extremes make this concrete: when the loss gradient concentrates on high-eigenvalue modes (large $p_k$), $\chi_{\text{pos}} \to 1$, indicating learning from strong, well-represented directions. Conversely, when the loss gradient lies in low-eigenvalue modes (small $p_k$), $\chi_{\text{pos}} \to 0$, signaling that the model exploits weak, fine-grained spectral directions to make progress.[2]

**From decomposition to efficiency.** While computing $\chi_{\text{pos}}$ would traditionally require an expensive full eNTK computation, our approach circumvents this limitation: the terms $\chi_{\text{net}}$ and $\chi_{\text{loss}}$ can be computed directly from sample-wise gradient magnitudes (Novak et al., 2019; Yousefpour et al., 2022). Additionally, we compute $\delta\mathcal{L}$, allowing us to access $\chi_{\text{pos}}$ indirectly via

$$\chi_{\text{pos}} = \delta\mathcal{L}/(\chi_{\text{net}} \cdot \chi_{\text{loss}}). \tag{10}$$

Unpacking $\delta\mathcal{L} = \|\nabla_{\boldsymbol{\theta}}\mathcal{L}\|^2$ shows that all three quantities ($\delta\mathcal{L}$, $\chi_{\text{net}}$, $\chi_{\text{loss}}$) are instantaneous properties of the network at the current parameters $\boldsymbol{\theta}_t$, obtained from gradient evaluations alone. The non-linear term $\nu$ and the step size $\eta$ do not enter, making this an exact computation of $\chi_{\text{pos}}$, not an approximation. Recasting $\chi_{\text{pos}}$ as a ratio of cheaply-computed quantities avoids the eNTK computation entirely. Our approach adds minimal overhead: computing the analysis in Figure 1 alongside training increases total compute and memory costs by less than $2\times$ (details on computational overhead are provided in Appendix B.1). This enables fine-grained analysis of learning dynamics at scales where explicit eNTK methods are impractical, including modern vision transformers and language models used in scaling-law research.

**In Practice** we use a slightly more general form of the loss decomposition provided in Equation 24. The full derivation is provided in Appendix A, where we take into account the nature of stochastic mini-batch training (Appendix A.1), and introduce correct batch-size scaling (Appendix A.2). The LNP-components retain their interpretations, but $\chi_{\text{pos}}$ can become negative due to stochastic effects and overfitting.

### 3.2. Verification

To validate our measurement framework, we evaluate the empirical LNP-components in a controlled setting where we can derive theoretical predictions. RFMs trained on Gaussian data with a power-law covariance spectrum provide such a setting (Maloney et al., 2022; Bordelon et al., 2025). We draw data from a Gaussian distribution with covariance eigenvalues that decay as a power law with exponent $\beta$, i.e., $\mu_k \propto k^{-\beta}$. We adopt a teacher-student framework (Canatar

et al., 2021; Bahri et al., 2024), where a teacher model generates samples that a student model attempts to learn. We ensure that the teacher and student share the same RKHS by sharing the random feature map. For more details, see Appendix B.2.5.

This allows us to control the data complexity through the spectral exponent $\beta$, visualized in Figure 3 (left). Specifically, larger $\beta$ values correspond to more concentrated spectra (lower complexity), while smaller $\beta$ values yield more spread-out spectra (higher complexity).

In Appendix A.4, we derive theoretical predictions for the scaling behavior of the LNP-components at initialization as a function of the spectral exponent $\beta$. In the limit of infinite-width RFMs and full-batch gradient descent, we find

$$\chi_{\text{loss}}(\beta) = \chi_{\text{net}}(\beta) = 1, \quad \chi_{\text{pos}}(\beta) \propto \frac{\sum_k k^{-2\beta}}{\left(\sum_k k^{-\beta}\right)^2}. \tag{11}$$

Figure 3 (right) empirically confirms this theoretical prediction. All three LNP-components match their predicted behavior: $\chi_{\text{net}}$ and $\chi_{\text{loss}}$ remain constant across different spectral exponents, while $\chi_{\text{pos}}$ precisely follows the predicted scaling relation, capturing the effect of data complexity. This validates both our measurement approach and the interpretation of $\chi_{\text{pos}}$ as a spectral complexity measure.

**From idealized to realistic settings.** While RFMs provide a controlled baseline for validation, realistic training introduces factors whose effects remain unclear: feature learning continuously reshapes the kernel, stochastic mini-batch updates inject noise into gradient directions, and complex data may not be well-represented in the initial kernel's RKHS. Without empirical measurements in these realistic regimes, we cannot determine whether or how these factors alter the scaling behavior predicted by idealized theory. This motivates the scalable LNP framework, which enables tracking spectral position dynamics in settings where scaling laws are actually observed.

## 4. Spectral Position and Scaling Laws

**Why do larger models learn better?** To address this question, we study the spectral position $\chi_{\text{pos}}$ throughout training in settings where scaling laws emerge. We train Llama 2 models (Touvron et al., 2023) of varying sizes on SimpleStories (Finke et al., 2025) and next-pixel prediction on CIFAR-5M (Nakkiran et al., 2020; Qiu et al., 2025) as these datasets allow us to observe scaling laws within computationally affordable regimes. We use the AdamW optimizer with warmup on single-pass training for both settings. Details are provided in Appendix B.2. Throughout training, we measure $\chi_{\text{pos}}$ using the LNP-decomposition. Figure 1 shows the observed scaling laws and spectral position dy-

---

[2]$\chi_{\text{pos}}$ has a precise mathematical relation to Cristianini's kernel-target alignment (Cristianini et al., 2001), differing in the eNTK normalization; see Appendix A.3.

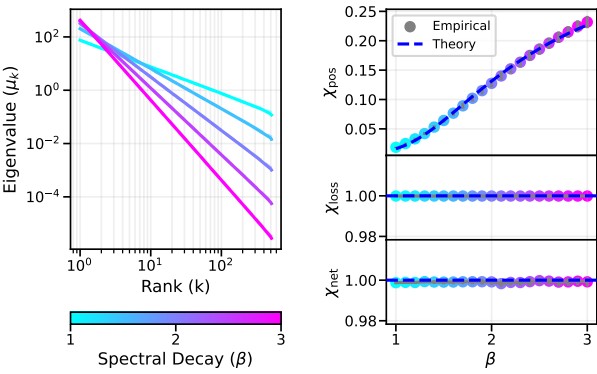

*Figure 3.* Validation of LNP-components on random feature models with power-law data spectra. **(left)** Covariance spectra of Gaussian data for varying spectral decay exponents $\beta \in [1.0, 3.0]$. Larger $\beta$ yields more concentrated (lower complexity) spectra. **(right)** Empirical and theoretical predictions of LNP-components $\chi_{\text{net}}$, $\chi_{\text{loss}}$, and $\chi_{\text{pos}}$ at model initialization as a function of $\beta$. All precisely follow the predicted scaling from Equation 11. Empirical values are averaged over 100 seeds; error bars are shown in grey.

namics throughout training. We observe clear scaling laws in both settings, with larger models achieving lower training losses. The spectral position shows two key phenomena:

1. Spectral position decreases throughout training over orders of magnitude.

2. Larger models reach lower spectral position values than smaller models.

In Appendix D.3, we provide additional results on a classification task using Vision Transformers trained on CIFAR-5M, confirming that the observed decrease in spectral position generalizes to vision classification settings.

**What does a decreasing spectral position tell us about the learning process?** The key to interpreting this observation lies in spectral bias: neural networks learn hierarchically, capturing dominant patterns before fine details (Rahaman et al., 2019; Canatar et al., 2021). In the NTK framework, this hierarchy maps onto the eigenspectrum: large-eigenvalue modes correspond to coarse, "easy" patterns that are learned fastest, while small-eigenvalue modes encode fine-grained details that require more training to resolve (Cao et al., 2020; Bordelon et al., 2020).

As training progresses, the dominant modes converge first: the model fits the broad structure of the data, and the prediction error in those directions vanishes. Once a mode has converged, it no longer contributes to the loss gradient, as there is simply no remaining error to correct in that direction. This is precisely what the spectral position captures: as noted in Section 3, $\chi_{\text{pos}}$ weights the eNTK spectrum by the projection coefficients $\gamma_k$, which quantify how much

the loss gradient projects onto each eigenmode. The observed decrease in $\chi_{\text{pos}}$ therefore reflects a systematic shift where the loss gradient becomes increasingly orthogonal to already-learned high-eigenvalue modes and the coefficients $\gamma_k$ migrate toward small-eigenvalue modes where residual error persists and learning continues.[3]

A low $\chi_{\text{pos}}$ is therefore not a sign of poor learning, but rather the *signature of a model that has successfully learned the coarse structure and is now refining fine-grained details*.

**How does model size affect a model's spectral reach?** We define **Spectral Reach** as a model's capacity to learn from progressively smaller eigenvalue modes of the eNTK spectrum. Spectral position is our measure for that, since lower $\chi_{\text{pos}}$ indicates that learning has progressed deeper into the spectral tail, where only fine-grained residual errors remain.

In Figure 1, we observe that smaller models "flatten out"—their spectral position stops decreasing even as training continues. We interpret this plateau as a capacity limit: the model has reached a floor beyond which $\chi_{\text{pos}}$ cannot decrease further, suggesting that finer-grained learning is no longer accessible. Larger models, by contrast, sustain the downward trajectory, reaching significantly lower spectral position values. We hypothesize that their richer parameter space gives rise to an eNTK spectrum with greater resolution in the small-eigenvalue regime: more eigenmodes that can capture fine-grained patterns.

To rule out width-dependent feature-learning intensity as a confound, we replicate our vision analysis in Appendix D.3 under maximal-update parameterization (muP) (Yang et al., 2021; 2022), which holds feature-learning intensity constant across widths. Figure 9 (right) demonstrates that the spectral-reach ordering across model sizes is preserved, confirming it as a capacity-driven effect.

**Connecting spectral reach to performance.** The concept of spectral reach has been observed in prior theoretical works on linear kernel models, where the tail of the kernel spectrum governs the residual error (Bordelon et al., 2020; Bahri et al., 2024): Small models lack the capability to resolve the tail, leaving corresponding target components unlearned. Our observations extend this picture to practical deep learning settings with empirical evidence. However, verifying this hypothesis crucially requires an efficient measure of *which* parts of the spectrum are utilized *at each stage* of learning—precisely what spectral position provides.

Summarizing, spectral reach provides a perspective on why

---

[3]We occasionally observe sudden drops in $\chi_{\text{pos}}$ beyond the smooth decrease predicted by spectral bias (Figure 1, right). We interpret these as *feature-learning events*: representation reorganizations that restructure the eNTK eigenvalue distribution and open access to previously unreachable spectral modes.

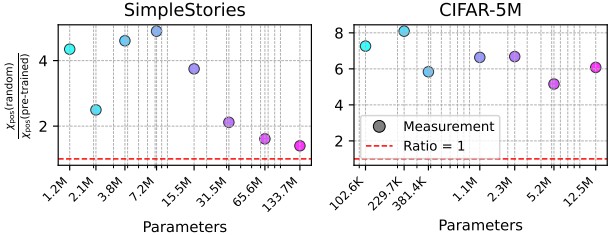

*Figure 4.* Effect of feature learning on spectral reach: Spectral position ratio $\frac{\chi_{\text{pos}}(\text{random})}{\chi_{\text{pos}}(\text{pre-trained})}$ of linear probing (only last layer trained, rest frozen) on pre-trained vs. random backbones for Llama 2 models of varying sizes. Shown are results on SimpleStories and CIFAR-5M datasets. We show averaged values over the last 10% of training to capture the steady-state behavior. Further details on the experimental setup are provided in Appendix B, and on the evaluation procedure in Appendix C.

larger models achieve lower loss: they can learn from weak spectral signals that are inaccessible to smaller models, enabling continued refinement and performance gains.

## 5. Feature Learning Enables Spectral Reach

### 5.1. Linear Probing to Connect to Feature Learning

Section 4 established that larger models achieve greater spectral reach, accessing fine-grained spectral modes that smaller models cannot. But what enables this capability? To isolate the role of feature learning, we compare linear probing on pre-trained versus random backbones. This comparison freezes all layers except the last, preventing feature adaptation during training and revealing whether learned representations inherently increase spectral reach.

**Hypothesis.** If feature learning enhances spectral reach, we expect that linear probes on pre-trained backbones will achieve lower spectral position values than those on random backbones. Formally, we hypothesize that

$$1 < \frac{\chi_{\text{pos}}(\text{random})}{\chi_{\text{pos}}(\text{pre-trained})} . \quad (12)$$

**Experimental setup.** We only train the final layer of Llama models on SimpleStories and CIFAR-5M, using the same training configuration as in Section 4. As backbones, we compare random initializations against pre-trained weights obtained from final checkpoints of the full training runs presented in Figure 1.

**Results.** Figure 4 presents the spectral position ratio from Equation 12, confirming our hypothesis across both datasets and all model sizes. Pre-trained backbones consistently reach lower spectral position values than random backbones. Notably, this difference emerges purely from the

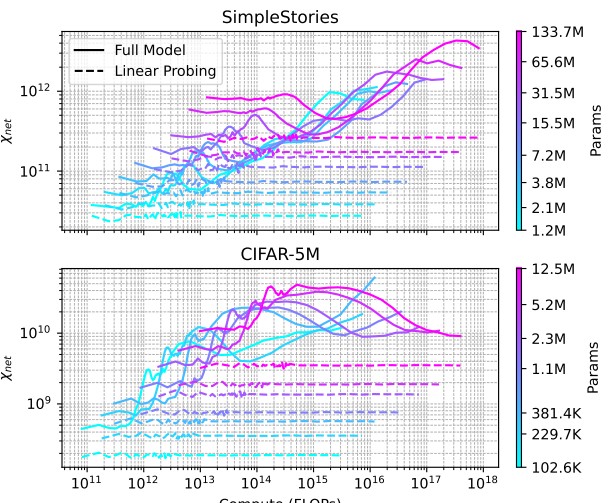

*Figure 5.* Training dynamics of gradient magnitudes $\chi_{\text{net}} = \|\nabla_\theta f\|_F^2$ for Llama models of varying sizes on SimpleStories and CIFAR-5M. We compare full training (all layers trained) to linear probing (only last layer trained, rest frozen) on a random backbone.

learned representations—neither backbone adapts during linear probing. This demonstrates that feature learning shapes the internal structure of representations, creating modes that support greater spectral reach.

### 5.2. How Feature Learning Affects Spectral Reach

Having established that feature learning enables greater spectral reach, we now ask: *how*? What mechanism allows feature-learning models to access spectral modes that frozen-representation models cannot?

**The factorization puzzle.** Section 4 reported that $\chi_{\text{pos}}$ drops by orders of magnitude throughout training (Figure 1). In the factorization $\delta\mathcal{L} = \chi_{\text{loss}} \cdot \chi_{\text{net}} \cdot \chi_{\text{pos}}$, a strong decrease in any one factor drags the loss change down with it, potentially stalling learning progress. Here we turn to the other two LNP-components, $\chi_{\text{loss}}$ and $\chi_{\text{net}}$, and how their dynamics shape the loss change.

$\chi_{\text{loss}}$, defined in Section 3, depends only on the loss function and declines monotonically as models improve, so it can only reinforce rather than counteract the potential stall.[4] The remaining question is how $\chi_{\text{net}}$ evolves and whether feature learning enables it to adapt in a way that helps sustain learning progress as $\chi_{\text{pos}}$ decreases. To address this, we compare the dynamics of $\chi_{\text{net}}$ between feature-learning models (all layers trainable) and linear probes (only last layer trainable) in Figure 5. We observe that feature-learning models show increasing $\chi_{\text{net}}$ throughout training, while linear probes on

---

[4]Empirically verified in Figures 10 and 11.

random backbones maintain constant $\chi_{\text{net}}$.

**Why $\chi_{\text{net}}$ evolves only under feature learning.** In models with frozen backbones (random feature models or linear probes with fixed backbones), the eNTK spectrum is fixed throughout training. Since $\chi_{\text{net}} = \text{Tr}(\Theta)$ equals the trace of the eNTK, a frozen spectrum directly implies constant $\chi_{\text{net}}$. In feature-learning models, by contrast, representations continuously change during training, reshaping the eNTK spectrum and letting $\chi_{\text{net}}$ evolve. Throughout training, $\chi_{\text{net}}$ increases, acting as a partial counterweight to the $\chi_{\text{pos}}$ decrease. While the $\chi_{\text{pos}}$ decrease still dominates the product $\chi_{\text{net}} \cdot \chi_{\text{pos}}$ (Figures 10 and 11), $\chi_{\text{net}}$ growth buffers roughly one order of magnitude of that decrease, which is a substantial contribution to sustaining learning progress as the model accesses finer-grained spectral modes.

**Primary and secondary dynamics.** Spectral bias theory establishes that NTK-regime learning progresses from dominant toward subordinate eigenmodes, and that learning subtle signals in the spectral tail requires the loss gradient to concentrate there, which is what a small $\chi_{\text{pos}}$ measures. We therefore read a low $\chi_{\text{pos}}$ as the primary requirement for spectral reach. The $\chi_{\text{net}}$ increase observed in feature-learning models (but not in frozen probes) we read as a secondary, complementary dynamic: representation adaptation reshapes the eNTK spectrum in a way that supports continued learning as the model progresses into the spectral tail. A definitive mechanistic account would require controlled interventions, which we discuss in Section 6.

## 6. Discussion

**Summary of contributions.** This work introduces the LNP-decomposition, factoring loss dynamics into three interpretable components: $\chi_{\text{net}}$, $\chi_{\text{loss}}$, and $\chi_{\text{pos}}$. By bypassing full eNTK computation while capturing its spectral dynamics, the decomposition enables analysis at transformer scales with less than $2\times$ overhead.

In controlled random-feature models, empirical spectral position precisely matches theory-predicted scaling relations, validating both measurement accuracy and the interpretation of $\chi_{\text{pos}}$ as a spectral complexity measure.

Applying the LNP-decomposition to Llama models on SimpleStories and CIFAR-5M reveals two key observations: spectral position decreases by orders of magnitude throughout training, and larger models reach lower values. We interpret this through *spectral reach*: the capacity to learn from progressively smaller eigenvalue modes of the eNTK spectrum. This suggests why larger models achieve lower loss: they sustain learning on weak spectral signals that smaller models cannot access.

Comparing pre-trained and random backbones in linear-probing experiments further reveals a complementary

dynamic: feature learning shapes representations so that $\chi_{\text{net}}$ grows as $\chi_{\text{pos}}$ decreases, buffering the slowdown that learning weak signals would otherwise impose as the model moves into the spectral tail.

The specific interventions this picture suggests—accelerating spectral descent or reshaping the spectrum—are implications we sketch in Section 7.

**Limitations.** Our analysis captures first-order effects of the loss change around the current parameters. The LNP-components themselves are exact instantaneous properties of the network (their definition and computation do not depend on this linearization), but the non-linear correction $\nu_t$ is not analyzed in this work, and the extent to which it carries additional information about training dynamics remains open.

Empirically we evaluate spectral reach on Llama models across SimpleStories and CIFAR-5M next-pixel prediction, plus vision transformers on CIFAR-5M classification with standard and muP parameterization. Broader validation across more tasks, architectures, and larger model sizes is needed to confirm generality.

As this work focuses on scaling behavior, we use well-tuned training configurations per model size rather than systematically varying training hyperparameters. The LNP-decomposition is also well suited to analyzing how the learning rate, including in the edge-of-stability regime, shapes the interplay between linear and non-linear contributions to learning; we leave this as a natural application for future work.

**Causal direction vs. mechanism.** Our results connect spectral position to model scale and performance, but the causal status of these relationships deserves a careful statement.

*Direction.* That spectral bias drives the $\chi_{\text{pos}}$ decrease, rather than the decrease being a side-effect of other dynamics, is supported by three converging lines of evidence: (i) spectral-bias theory predicts exactly this trajectory (Rahaman et al., 2019; Cao et al., 2020; Bordelon et al., 2020), since $\chi_{\text{pos}}$ measures the eigenvalue at which learning is currently focused; (ii) the scale ordering persists under muP parameterization, which holds feature-learning intensity constant across widths, ruling out width-dependent feature learning as the driver (Section 4, Appendix D.3); (iii) the $\chi_{\text{pos}}$ decrease substantially exceeds the $\chi_{\text{net}}$ increase across training (Section 5, Appendix D.4), so $\chi_{\text{pos}}$ cannot be passively explained by trace inflation.

*Mechanism.* A definitive account of the mechanism involves understanding how feature learning restructures the eNTK spectrum to enable further spectral descent, and under what conditions it does not. This would require controlled interventions (for example, clamping $\chi_{\text{pos}}$ or targeted spectral reshaping). We view such interventions as a natural next

step, and outline two candidate directions in Section 7.

## 7. Outlook

**Methods to enhance spectral reach.** Our results reframe the optimization question: "how do we reduce loss?" becomes "how do we enhance spectral reach?", which opens two complementary intervention strategies.
*Accelerating spectral descent.* Interventions acting on the optimizer side, such as targeted learning-rate schedules, gradient scaling, or $\chi_{\text{net}}$ boosting, could push the model faster into subordinate spectral modes of the eNTK spectrum. How far this can be pushed without destabilizing training is an empirical question, as normalization layers and related mechanisms set limits on how much $\chi_{\text{net}}$ can grow.
*Reshaping the spectrum.* Interventions acting on the representations themselves, such as architectural choices or initialization schemes like muP (Yang et al., 2021), He, or Xavier (Glorot & Bengio, 2010), could compress the spectral tail toward the bulk, making subordinate modes easier to reach in the first place.

**Toward mode-level analysis of semantic structure.** Learning hierarchical data proceeds bottom-up, where low-level features (tokens forming words) are learned before high-level ones (complex structures like sentences or paragraphs) (Cagnetta et al., 2024). An intriguing question is whether spectral modes correspond to these hierarchical abstraction levels. If so, spectral reach would directly relate to the depth of semantic understanding a model achieves, and could serve as a quantitative measure of model capability. Resolving this question extends our work, since $\chi_{\text{pos}}$ already provides an efficient diagnostic. If $\chi_{\text{pos}}$ were evaluated on inputs whose loss gradient concentrates on a specific eNTK eigenmode, the aggregate would become a mode-resolved diagnostic that reports when the network begins and ends to learn that mode. The picture is analogous to band-structure spectroscopy: the bands theoretically exist, but resolving them requires probes that selectively excite each one. Constructing such mode-isolating inputs without first materializing the full eigendecomposition is the open problem, and solving it would complete the bridge from spectral reach to the semantic content of what the network learns.

**Beyond pre-training: signatures of paradigm transitions.** The scale-free nature of $\chi_{\text{pos}}$ makes it a natural diagnostic for transitions between training paradigms, where the loss function, and therefore its scale, changes. A concrete example is the transition from supervised fine-tuning to preference optimization methods such as DPO (Rafailov et al., 2023): the decision of when to switch is currently based on heuristics such as validation plateaus or fixed step budgets, whereas spectral reach offers a potentially principled signal.

A working hypothesis is that paradigm shifts move modes from the tail of the previous objective back toward the bulk under the new one, regenerating spectral reach for continued learning. Tracking $\chi_{\text{pos}}$ across transitions could therefore provide a quantitative handle on when and why paradigm changes are beneficial.

## Software and Data

All code, plots, and measurement artifacts are available at (Nikolaou et al., 2026b). Experiments were run with PyTorch (Paszke et al., 2019); the LNP-decomposition is implemented in the open-source `perspic` package (Nikolaou & Scheunemann, 2026). Training data are SimpleStories (Finke et al., 2025) and CIFAR-5M (Nakkiran et al., 2020).

## Acknowledgements

We thank the Deutsche Forschungsgemeinschaft (DFG, German Research Foundation) for supporting this work by funding—EXC2075—390740016 under Germany's Excellence Strategy. We acknowledge the support by the Stuttgart Center for Simulation Science (SimTech). We acknowledge funding from the Deutsche Forschungsgemeinschaft (DFG, German Research Foundation) through the Compute Cluster Grant no. 492175459. The authors acknowledge support by the state of Baden-Württemberg through bwHPC and the German Research Foundation (DFG) through grant INST 35/1597-1 FUGG and grant 455787709. CH and ST acknowledge financial support from the German Funding Agency (Deutsche Forschungsgemeinschaft DFG) under the Priority Program SPP 2363, 'Utilization and Development of Machine Learning for Molecular Applications-Molecular Machine Learning' Project no. 497249646. SK's work has been partially supported by STFC consolidated Grants ST/T000694/1 and ST/X000664/1. The authors thank the International Max Planck Research School for Intelligent Systems (IMPRS-IS) for supporting Konstantin Nikolaou. KN is grateful for the kind hospitality of the Department of Applied Mathematics and Theoretical Physics, University of Cambridge, where part of this work was carried out.

## Impact Statement

This paper presents work whose goal is to advance the field of Machine Learning. There are many potential societal consequences of our work, none which we feel must be specifically highlighted here.

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

# A. Derivations

## A.1. Derivation of LNP-Components

In this section, we provide a detailed derivation of the decomposition of loss dynamics into three components: the loss-network-position (LNP) decomposition. This framework enables computationally efficient calculation of the spectral position $\chi_{\mathrm{pos}}$, which typically would require expensive computations of the empirical neural tangent kernel (eNTK). While the main idea is presented in Section 3, we here provide a detailed derivation of the framework that includes dynamics of stochastic mini-batch training.

Let $\mathbf{X} = [\mathbf{x}_1, ..., \mathbf{x}_D]^T$ denote the matrix of $D$ training inputs $\mathbf{x}_i \in \mathbb{R}^n$ and $\mathbf{Y} = [y_1, ..., y_D]^T$ be the corresponding matrix of targets $\mathbf{y}_i \in \mathbb{R}^m$, which can be either supervised labels or self-supervised targets. Let the neural network be represented by a function $f(\mathbf{x}; \boldsymbol{\theta}) \in \mathbb{R}^m$ with $N$ parameters $\boldsymbol{\theta} \in \mathbb{R}^N$.

The training process involves minimizing a loss function $\mathcal{L}(f(\mathbf{X}; \boldsymbol{\theta}), \mathbf{Y}) =: \mathcal{L}(\mathbf{X}, \boldsymbol{\theta})$, over the dataset. For simplicity, we omit the explicit notation of targets, as each input has a corresponding pair. During training, the parameters $\boldsymbol{\theta}_t$ at training step $t \in \mathbb{N}$ are updated iteratively using stochastic gradient descent according to

$$\Delta\boldsymbol{\theta}_t := \boldsymbol{\theta}_{t+1} - \boldsymbol{\theta}_t = -\eta\nabla_{\boldsymbol{\theta}}\mathcal{L}(\mathbf{X}_t, \boldsymbol{\theta}_t), \tag{13}$$

where $\eta$ is the learning rate and $\mathbf{X}_t$ is the mini-batch of data sampled at training step $t$. In each training step, the loss changes according to

$$\Delta\mathcal{L}_t := \mathcal{L}(\mathbf{X}_{t+1}, \boldsymbol{\theta}_{t+1}) - \mathcal{L}(\mathbf{X}_t, \boldsymbol{\theta}_t), \tag{14}$$

where the updated loss is typically evaluated on a new mini-batch $\mathbf{X}_{t+1}$. Analyzing the full non-linear dynamics of neural networks during training is challenging, which is why we will focus on the linear component of the loss change. That is, for small step sizes $\eta$, we can linearize the loss change around the current parameters $\theta_t$:

$$\Delta\mathcal{L}_t = \underbrace{\nabla_{\boldsymbol{\theta}}^T\mathcal{L}(\mathbf{X}_{t+1}, \boldsymbol{\theta}_t)\,\Delta\boldsymbol{\theta}_t}_{\text{(linearized change)}} + \underbrace{O(\eta^2)}_{=:\,\nu_t\text{ (non-linear correction)}} \tag{15}$$
$$= -\eta\,\nabla_{\boldsymbol{\theta}}^T\mathcal{L}(\mathbf{X}_{t+1}, \boldsymbol{\theta}_t)\,\nabla_{\boldsymbol{\theta}}\mathcal{L}(\mathbf{X}_t, \boldsymbol{\theta}_t) + \nu_t$$

Note that the loss change is induced by the mini-batch $\mathbf{X}_t$ sampled at step $t$, while the loss is evaluated on a different mini-batch $\mathbf{X}_{t+1}$ at step $t + 1$. This is an extension of the derivation provided in Section 3, where we assumed full-batch training for simplicity.

We will now focus on the linearized loss change term in Equation 15 and derive its LNP-decomposition. To do this, we first simplify the notation: we drop the explicit dependence of the parameters $\boldsymbol{\theta}_t$ and abstract the mini-batches $\mathbf{X}_t$ and $\mathbf{X}_{t+1}$ as batches $\mathbf{X}_A$ and $\mathbf{X}_B$, respectively. This allows us to express the linearized loss change more compactly as

$$\delta\mathcal{L}(A, B) := \nabla_{\boldsymbol{\theta}}^T\mathcal{L}^B\,\nabla_{\boldsymbol{\theta}}\mathcal{L}^A := \nabla_{\boldsymbol{\theta}}^T\mathcal{L}(\mathbf{X}_B, \boldsymbol{\theta})\,\nabla_{\boldsymbol{\theta}}\mathcal{L}(\mathbf{X}_A, \boldsymbol{\theta}) \tag{16}$$

We apply the chain rule to express the loss gradients with respect to the parameters $\boldsymbol{\theta}$ in terms of the network outputs $f$. For batch $A$, this reads

$$\nabla_{\boldsymbol{\theta}}\mathcal{L}^A = (\nabla_{\boldsymbol{\theta}}f^A)^T\,\nabla_f\mathcal{L}^A, \tag{17}$$

where $\nabla_f\mathcal{L} \in \mathbb{R}^{m|A|}$ is the loss gradient with respect to the network outputs and $\nabla_{\boldsymbol{\theta}}f \in \mathbb{R}^{N \times m|A|}$ is the Jacobian of the network outputs with respect to the parameters evaluated on batch $A$. $|A|$ and $|B|$ denote the batch sizes of batches $A$ and $B$, respectively. Note that we have vectorized the network outputs over the batch dimension, such that $f^A = [f(\mathbf{x}_i; \boldsymbol{\theta})]_{\mathbf{x}_i \in \mathbf{X}_A} \in \mathbb{R}^{m|A|}$ and $\nabla_f\mathcal{L}^A = [\nabla_{f(\mathbf{x}_i; \boldsymbol{\theta})}\mathcal{L}(\mathbf{X}_A, \boldsymbol{\theta})]_{\mathbf{x}_i \in \mathbf{X}_A} \in \mathbb{R}^{m|A|}$. Dataset $B$ is treated analogously. Inserting this into Equation 16, we obtain

$$\delta\mathcal{L}(A, B) = (\nabla_f\mathcal{L}^B)^T\underbrace{\nabla_{\boldsymbol{\theta}}f^B(\nabla_{\boldsymbol{\theta}}f^A)^T}_{=:\,\Theta^{AB}}\nabla_f\mathcal{L}^A, \tag{18}$$

where we have defined the empirical NTK between batches $A$ and $B$ as $\Theta^{AB} = \nabla_{\boldsymbol{\theta}}f^B(\nabla_{\boldsymbol{\theta}}f^A)^T \in \mathbb{R}^{m|B|\times m|A|}$. It is important to note that for mini-batch training with $A \neq B$, the matrix $\Theta^{AB}$ is not necessarily symmetric or positive

semi-definite, in contrast to the full-batch empirical NTK with $A = B$. To proceed, we perform an eigendecomposition onto the basis of $\Theta^{AA}$ and $\Theta^{BB}$:

$$\delta\mathcal{L}(A, B) = \sum_{k,l}(\nabla_f^T\mathcal{L}^B q_k^B)\left[(q_k^B)^T \Theta^{AB} q_l^A\right](\nabla_f^T\mathcal{L}^A q_l^A) \tag{19}$$

where $q_k^A$ and $q_k^B$ are the eigenvectors of $\Theta^{AA}$ and $\Theta^{BB}$, respectively.

**LNP-decomposition for Full-Batch Training.** Equation 19 generalizes Equation 6 to mini-batch training. This can be seen by noting that for full-batch training with $A = B$, the matrix $\Theta^{AB}$ becomes symmetric and positive semi-definite $\Theta := \Theta^{AA} = \Theta^{BB}$, allowing us to choose the same eigenbasis for both sides, i.e., $q_k^A = q_k^B = q_k$. The kernel then projects onto its eigenmodes $q_k^T \Theta q_l = \lambda_k \delta_{kl}$, yielding Equation 6:

$$\delta\mathcal{L} = \sum_k(\nabla_f^T\mathcal{L} q_k)^2 \lambda_k \tag{20}$$

In that case, the eigenvectors $q_k$ form a complete orthonormal basis, and the projections of the loss gradient onto this basis satisfy

$$\sum_k(\nabla_f^T\mathcal{L} q_k)^2 = \|\nabla_f\mathcal{L}\|_2^2 \ , \tag{21}$$

which is the squared magnitude of the loss gradient with respect to the network outputs and identifies the scale contribution of the loss function. Similarly, the eigenvalues $\lambda_k$ satisfy

$$\sum_k \lambda_k = \text{Tr}(\Theta) = \|\nabla_{\boldsymbol{\theta}} f\|_F^2 \tag{22}$$

where $\|\nabla_{\boldsymbol{\theta}} f\|_F^2$ is the squared Frobenius norm of the Jacobian of the network outputs with respect to the parameters. This term captures the overall sensitivity of the network outputs to parameter changes, and identifies the scale contribution of the network. Using these relations, we can factorize both scale contributions from the sum in Equation 6 to obtain the LNP-decomposition:

$$\delta\mathcal{L} = \underbrace{\|\nabla_f\mathcal{L}\|_2^2}_{=:\chi_{\text{loss}}} \underbrace{\|\nabla_{\boldsymbol{\theta}} f\|_F^2}_{=:\chi_{\text{net}}} \underbrace{\sum_k \frac{(\nabla_f^T\mathcal{L} q_k)^2}{\|\nabla_f\mathcal{L}\|_2^2} \frac{\lambda_k}{\|\nabla_{\boldsymbol{\theta}} f\|_F^2}}_{=:\chi_{\text{pos}}} = \chi_{\text{loss}} \cdot \chi_{\text{net}} \cdot \chi_{\text{pos}} \ , \tag{23}$$

Factorizing out the two scale contributions yields a third component $\chi_{\text{pos}}$ that captures the interaction between the network and the loss function through the eNTK's spectral properties. This component is scale-free and takes values in the range $\chi_{\text{pos}} \in [0, 1]$. For more information on the interpretation of the LNP-components, we refer to Section 3.

**LNP-decomposition for Mini-Batch Training.** For mini-batch training, the parameter update is carried out on batch $A$ and the loss change is evaluated on batch $B$ with $A \neq B$. We proceed similarly by factorizing the scale contributions from Equation 19:

$$\delta\mathcal{L}(A, B) = \underbrace{\|\nabla_f\mathcal{L}^B\|_2 \|\nabla_f\mathcal{L}^A\|_2}_{=:\chi_{\text{loss}}^{AB}} \underbrace{\|\nabla_{\boldsymbol{\theta}} f^B\|_F \|\nabla_{\boldsymbol{\theta}} f^A\|_F}_{=:\chi_{\text{net}}^{AB}} \underbrace{\sum_{k,l} \frac{(\nabla_f^T\mathcal{L}^B q_k^B)}{\|\nabla_f\mathcal{L}^B\|_2} \frac{(\nabla_f^T\mathcal{L}^A q_l^A)}{\|\nabla_f\mathcal{L}^A\|_2} \frac{(q_k^B)^T \Theta^{AB} q_l^A}{\|\nabla_{\boldsymbol{\theta}} f^B\|_F \|\nabla_{\boldsymbol{\theta}} f^A\|_F}}_{=:\chi_{\text{pos}}^{AB}}$$

$$= \chi_{\text{loss}}^{AB} \cdot \chi_{\text{net}}^{AB} \cdot \chi_{\text{pos}}^{AB} \ . \tag{24}$$

This expression generalizes the LNP-decomposition to mini-batch training, where each component now depends on both batches $A$ and $B$. We obtain a contribution of loss and network magnitude from both batches and a spectral position component $\chi_{\text{pos}}^{AB}$ that captures the interaction between the two batches through the mixed eNTK $\Theta^{AB}$. Note that the projections of the loss onto the eigenbases $\nabla_f^T\mathcal{L}^B q_k^B$ and $\nabla_f^T\mathcal{L}^A q_l^A$ can now result in negative values. Similarly, the mixed

eNTK projections $(q_k^B)^T \Theta^{AB} q_l^A$ can also be negative. If the spectral position contribution $\chi_{\text{pos }kl}^{AB}$ of a specific mode pair $(k,l)$ satisfies

$$\chi_{\text{pos }kl}^{AB} = \frac{(\nabla_f^T \mathcal{L}^B q_k^B)}{\|\nabla_f \mathcal{L}^B\|_2} \frac{(\nabla_f^T \mathcal{L}^A q_l^A)}{\|\nabla_f \mathcal{L}^A\|_2} \frac{(q_k^B)^T \Theta^{AB} q_l^A}{\|\nabla_\theta f^B\|_F \|\nabla_\theta f^A\|_F} < 0, \tag{25}$$

it means that the mode $l$ resulting from training on $A$ has a detrimental effect on the mode $k$ in $B$. This happens when information learned from batch $A$ does not generalize to batch $B$ or even interferes destructively, which occurs e.g. in unlearning and overfitting scenarios.

As this work is focused on pre-training settings with large and diverse datasets, we expect most mode pairs to contribute positively to the spectral position, i.e., $\chi_{\text{pos }kl}^{AB} > 0$. Comparing Figures 1 and 7, we see that train and test losses yield the same quantitative scaling ratios. This means that training seems to converge before overfitting effects occur, which supports our assumption of positive spectral position contributions.

### A.2. Normalization of LNP-Components

In this section, we provide further details how to normalize the LNP-components introduced in Section 3 and Appendix A.1 to remove batch-size dependencies.

The LNP-decomposition factors the linearized loss change into three components

$$\delta\mathcal{L}(A, B) = \chi_{\text{loss}}^{AB} \cdot \chi_{\text{net}}^{AB} \cdot \chi_{\text{pos}}^{AB}. \tag{26}$$

However, with the definition provided in Equation 24, the loss and network magnitudes $\chi_{\text{loss}}^{AB}$ and $\chi_{\text{net}}^{AB}$ depend on the batch sizes $|A|$ and $|B|$. This can be understood by rewriting into sample-wise contributions. For a loss function of the form $\mathcal{L}^B = \frac{1}{|B|} \sum_{i \in B} \mathcal{L}(f(\mathbf{x}_i), y_i)$, we can express the loss magnitude component $\|\nabla_f \mathcal{L}^B\|_2$ as sample-wise sum

$$\|\nabla_f \mathcal{L}^B\|_2 = \sqrt{\frac{1}{|B|^2} \sum_{i \in B} \|\nabla_{f(\mathbf{x}_i)}\mathcal{L}\|_2^2} = \frac{1}{\sqrt{|B|}} \sqrt{\frac{1}{|B|} \sum_{i \in B} \|\nabla_{f(\mathbf{x}_i)}\mathcal{L}\|_2^2} = \frac{1}{\sqrt{|B|}} \sigma_{\text{loss}}^B, \tag{27}$$

where $\sigma_{\text{loss}}^B$ is the standard deviation of the sample-wise loss gradients in batch $B$. In expectation of large batch sizes, this quantity converges to the population standard deviation of the loss gradients over the data distribution. Thus, the loss magnitude scales with the inverse square root of the batch size $\|\nabla_f \mathcal{L}^B\|_2 \propto \frac{1}{\sqrt{|B|}}$.

Looking at the network magnitude $\|\nabla_\theta f^B\|_F$, we can perform a similar derivation:

$$\|\nabla_\theta f^B\|_F = \sqrt{\sum_{i \in B} \|\nabla_\theta f(\mathbf{x}_i)\|_2^2} = \sqrt{|B|} \sqrt{\frac{1}{|B|} \sum_{i \in B} \|\nabla_\theta f(\mathbf{x}_i)\|_2^2} = \sqrt{|B|} \sigma_{\text{net}}^B, \tag{28}$$

where $\sigma_{\text{net}}^B$ is the standard deviation of the sample-wise network gradients in batch $B$. In expectation of large batch sizes, this quantity converges to the population standard deviation of the network gradients over the data distribution. Thus, the network magnitude scales with the square root of the batch size $\|\nabla_\theta f^B\|_F \propto \sqrt{|B|}$.

To remove these batch-size dependencies, we can move the batch-size scaling factors from the loss into the network magnitude, yielding normalized LNP-components:

$$\delta\mathcal{L}(A, B) = \underbrace{\sqrt{|A||B|}\, \chi_{\text{loss}}^{AB}}_{=:\, \tilde{\chi}_{\text{loss}}^{AB}} \underbrace{\frac{\chi_{\text{net}}^{AB}}{\sqrt{|A||B|}}}_{=:\, \tilde{\chi}_{\text{net}}^{AB}} \chi_{\text{pos}}^{AB} \tag{29}$$

In practice, we use these normalized LNP-components $\tilde{\chi}_{\text{loss}}^{AB}$ and $\tilde{\chi}_{\text{net}}^{AB}$ to analyze training dynamics across different batch sizes.

### A.3. Connection to Kernel-Target Alignment

$\chi_{\text{pos}}$ indicates which eigenvalues of the eNTK currently drive loss reduction (Section 3). Since this involves the eNTK and the loss-gradient direction together, a natural question is how it relates to kernel-alignment metrics. The kernel-target

alignment of Cristianini et al. (2001) and its follow-ups (Cortes et al., 2012; Baratin et al., 2021) provide a general framework for the affinity between two positive semi-definite matrices, typically a kernel and a task-defined target. $\chi_{\text{pos}}$ corresponds to a specific instance of this idea (loss gradient as target, eNTK as kernel) with a different normalization, which arises naturally from the LNP-decomposition. Below we make the connection explicit.

**$\chi_{\text{pos}}$ as a Matrix-Alignment Ratio.**   With slight abuse of notation, $A$ and $B$ in this subsection denote the two matrices being compared, not the batch indices used in Appendices A.1 and A.2. Define

$$A = \nabla_f \mathcal{L} \, (\nabla_f \mathcal{L})^T \in \mathbb{R}^{nm \times nm}, \tag{30}$$

$$B = \Theta \in \mathbb{R}^{nm \times nm}, \tag{31}$$

where $n$ is the batch size and $m$ the per-sample output dimension. Both $A$ and $B$ are positive semi-definite, and $A$ is rank-1 by construction: it is the outer product of $g = \nabla_f \mathcal{L} \in \mathbb{R}^{nm}$ with itself.

Recall from Section 3 that $\delta\mathcal{L} = (\nabla_f \mathcal{L})^T \Theta \nabla_f \mathcal{L} = \text{Tr}(AB)$, $\chi_{\text{loss}} = \|\nabla_f \mathcal{L}\|_2^2 = \text{Tr}(A)$, and $\chi_{\text{net}} = \|\nabla_\theta f\|_F^2 = \text{Tr}(\Theta) = \text{Tr}(B)$. Substituting these identities into the LNP-decomposition,

$$\chi_{\text{pos}} = \frac{\delta\mathcal{L}}{\chi_{\text{loss}} \cdot \chi_{\text{net}}} = \frac{\text{Tr}(AB)}{\text{Tr}(A) \cdot \text{Tr}(B)}. \tag{32}$$

$\chi_{\text{pos}}$ is therefore a ratio between the matrix overlap $\text{Tr}(AB)$ and the product of the two traces.

**Relation to Cristianini's Kernel-Target Alignment.**   Cristianini's measure normalizes by Frobenius norms instead of traces:

$$\text{Sim}(A, B) = \frac{\text{Tr}(AB)}{\|A\|_F \|B\|_F} = \frac{\text{Tr}(AB)}{\sqrt{\text{Tr}(A^2)} \cdot \sqrt{\text{Tr}(B^2)}}. \tag{33}$$

The two measures share the numerator $\text{Tr}(AB)$ and differ only in their denominators. The loss-side denominator simplifies because $A$ is rank-1: with $A = gg^T$, the matrix product reduces to $A^2 = g(g^T g)g^T = \|g\|^2 A$. Taking the trace,

$$\text{Tr}(A^2) = \|g\|^2 \, \text{Tr}(A) = \|g\|^4 = \text{Tr}(A)^2, \tag{34}$$

so $\sqrt{\text{Tr}(A^2)} = \text{Tr}(A)$. The two denominators therefore agree on the loss side; the only difference lies in how $\Theta$ is normalized: $\chi_{\text{pos}}$ uses $\text{Tr}(\Theta)$, while $\text{Sim}(A, B)$ uses $\|\Theta\|_F = \sqrt{\text{Tr}(\Theta^2)}$. The exact relationship is

$$\chi_{\text{pos}} = \text{Sim}(A, B) \cdot \frac{\|\Theta\|_F}{\text{Tr}(\Theta)}. \tag{35}$$

**Spectral Interpretation of the Extra Factor.**   Writing $\Theta = \sum_k \lambda_k \, q_k q_k^T$ with eigenvalues $\lambda_k \geq 0$, we have $\text{Tr}(\Theta) = \sum_k \lambda_k$ and $\|\Theta\|_F^2 = \sum_k \lambda_k^2$. By the Cauchy-Schwarz inequality $(\sum_k \lambda_k)^2 \leq nm \sum_k \lambda_k^2$, so

$$\frac{\|\Theta\|_F}{\text{Tr}(\Theta)} \in \left[ \frac{1}{\sqrt{nm}}, 1 \right], \tag{36}$$

with the lower bound attained when all eigenvalues are equal (maximally spread spectrum) and the upper bound when one eigenvalue dominates (maximally concentrated spectrum). Equivalently, with normalized eigenvalues $p_k = \lambda_k / \text{Tr}(\Theta)$, the eNTK's inverse participation ratio is $\text{IPR}(\Theta) = \sum_k p_k^2 = \|\Theta\|_F^2 / \text{Tr}(\Theta)^2$, so

$$\frac{\|\Theta\|_F}{\text{Tr}(\Theta)} = \sqrt{\text{IPR}(\Theta)}. \tag{37}$$

The factor measures spectral concentration of the eNTK: small when eigenvalues are spread out, large when they cluster on a small effective subspace.

Spectral position therefore jointly captures the directional alignment between the loss gradient and the eNTK eigenmodes *and* the spectral concentration of the eNTK itself. The trace-based normalization is not arbitrary: it arises from the LNP factorization derived in Section 3 and is what guarantees the bounded range $\chi_{\text{pos}} \in [0, 1]$.

### A.4. Derivation of Spectral Scaling Relationships for RFMs

This section derives the theoretical scaling relationship between the data spectral exponent $\beta$ and the loss-network-position (LNP) decomposition components. We analyze a teacher-student setting for random feature models (RFMs) in the infinite-width limit.

**Notation note.** This subsection and the corresponding experimental details in Appendix B.2.5 adopt standard random-feature / kernel-methods conventions: $n$ denotes the dataset size, $m$ the width, $\mathbf{w}_j$ a frozen random feature, and $\mathbf{c}$ the trainable readout weights. These differ from the NTK notation in Appendices A.1–A.3.

A.4.1. SETUP AND DEFINITIONS

We consider a dataset $\mathcal{D} = \{(\mathbf{x}_i, y_i)\}_{i=1}^n$ where inputs $\mathbf{x} \in \mathbb{R}^d$ are drawn i.i.d. from a centered Gaussian distribution $\mathcal{N}(0, \boldsymbol{\Sigma})$ with trace normalized to $d$. The model is a random feature model in the infinite-width limit ($m \to \infty$), which is equivalent to a kernel regression method with the limiting kernel:

$$K(\mathbf{x}, \mathbf{x}') = \mathbb{E}_{\mathbf{w} \sim \mathcal{N}(0, d^{-1}\mathbf{I})} \left[ \sigma(\mathbf{w}^\top \mathbf{x}) \sigma(\mathbf{w}^\top \mathbf{x}') \right], \tag{38}$$

where $\sigma(\cdot)$ is the variance-preserving ReLU activation $\sigma(z) = \sqrt{2} \max(0, z)$. We denote the kernel Gram matrix as $\mathbf{K} \in \mathbb{R}^{n \times n}$ with entries $K_{ij} = K(\mathbf{x}_i, \mathbf{x}_j)$. The network is trained to minimize the Mean Squared Error (MSE):

$$\mathcal{L}(\mathbf{c}) = \frac{1}{2n} \|\mathbf{y} - f(\mathbf{X}; \mathbf{c})\|^2. \tag{39}$$

A.4.2. LOSS DYNAMICS AT INITIALIZATION

We analyze the change in loss $\Delta\mathcal{L}$ after a single gradient descent step with learning rate $\eta$, starting from zero initialization ($\mathbf{c}_0 = \mathbf{0}$). The network output after one step is $f_1(\mathbf{X}) = \frac{\eta}{n}\mathbf{K}\mathbf{y}$. The exact change in loss is:

$$\begin{aligned} \Delta\mathcal{L} &= \mathcal{L}(\mathbf{c}_1) - \mathcal{L}(\mathbf{0}) \\ &= \frac{1}{2n} \left\| \mathbf{y} - \frac{\eta}{n}\mathbf{K}\mathbf{y} \right\|^2 - \frac{1}{2n}\|\mathbf{y}\|^2 \\ &= -\frac{\eta}{n^2}\mathbf{y}^\top\mathbf{K}\mathbf{y} + \frac{\eta^2}{2n^3}\mathbf{y}^\top\mathbf{K}^2\mathbf{y}. \end{aligned} \tag{40}$$

In the small learning rate regime, the linear term dominates. We focus on the normalized initial slope of the loss:

$$-\frac{\Delta\mathcal{L}}{\eta} \approx \frac{1}{n^2}\mathbf{y}^\top\mathbf{K}\mathbf{y}. \tag{41}$$

A.4.3. LNP-DECOMPOSITION

We decompose the driving term $\frac{1}{n^2}\mathbf{y}^\top\mathbf{K}\mathbf{y}$ into the three scalar components: loss magnitude $\chi_{\text{loss}}$, network magnitude $\chi_{\text{net}}$, and spectral position $\chi_{\text{pos}}$. Using the eigendecomposition $\mathbf{K} = \sum_k \hat{\lambda}_k \mathbf{v}_k \mathbf{v}_k^\top$ and projecting the targets onto the eigenbasis as $\alpha_k = \mathbf{v}_k^\top \mathbf{y}$, we factorize as follows:

$$\frac{\mathbf{y}^\top\mathbf{K}\mathbf{y}}{n^2} = \underbrace{\left(\frac{\|\mathbf{y}\|^2}{n}\right)}_{\chi_{\text{loss}}} \cdot \underbrace{\left(\frac{\text{Tr}(\mathbf{K})}{n}\right)}_{\chi_{\text{net}}} \cdot \underbrace{\left(\sum_{k=1}^n \frac{\alpha_k^2}{\|\mathbf{y}\|^2} \frac{\hat{\lambda}_k}{\text{Tr}(\mathbf{K})}\right)}_{\chi_{\text{pos}}}. \tag{42}$$

A.4.4. COMPONENT ANALYSIS IN THE LARGE SAMPLE LIMIT

We evaluate the expectation of each component over the data and teacher distributions in the large $n$ limit.

**1. Loss Component ($\chi_{\text{loss}}$).** We construct targets to be standardized such that $\text{var}(y) = 1$. Implementation details can be found in Section B.2.5. Consequently, by the Law of Large Numbers:

$$\chi_{\text{loss}} = \frac{1}{n} \sum_{i=1}^n y_i^2 \xrightarrow{n \to \infty} \mathbb{E}[y^2] = 1. \tag{43}$$

**2. Network Component ($\chi_{\mathbf{net}}$).** The trace term averages the kernel diagonal: $\chi_{\text{net}} = \frac{1}{n} \sum_i K(\mathbf{x}_i, \mathbf{x}_i)$. For ReLU features with normalized weights $\mathbf{w} \sim \mathcal{N}(0, d^{-1}\mathbf{I})$ and data covariance $\text{Tr}(\boldsymbol{\Sigma}) = d$:

$$
\begin{aligned}
\mathbb{E}[\chi_{\text{net}}] &= \mathbb{E}_{\mathbf{x},\mathbf{w}}[\sigma(\mathbf{w}^\top \mathbf{x})^2] \\
&= \frac{1}{2}\mathbb{E}[(\mathbf{w}^\top \mathbf{x})^2] \cdot 2 \quad \text{(symmetry of distribution around 0)} \\
&= \text{Tr}\left(\mathbb{E}[\mathbf{x}\mathbf{x}^\top]\mathbb{E}[\mathbf{w}\mathbf{w}^\top]\right) = \text{Tr}\left(\boldsymbol{\Sigma} \cdot \frac{1}{d}\mathbf{I}\right) = 1.
\end{aligned}
\tag{44}
$$

Thus, $\chi_{\text{net}} \to 1$.

**3. Spectral Position Component ($\chi_{\mathbf{pos}}$).** This component captures how the target energy is distributed across the kernel's eigenmodes. We adopt a *Gaussian Process Teacher* setting, where $f^* \sim \mathcal{GP}(0, K)$. This implies that the vector of targets on the training set follows $\mathbf{y} \sim \mathcal{N}(0, \mathbf{K})$. In the eigenbasis of $\mathbf{K}$, the projection coefficients $\alpha_k$ are independent Gaussian variables with variance equal to the eigenvalues:

$$
\mathbb{E}[\alpha_k^2] = \hat{\lambda}_k.
\tag{45}
$$

Substituting this into the definition of $\chi_{\text{pos}}$ and invoking the concentration of spectral sums for large $n$ (replacing empirical $\hat{\lambda}$ with population $\lambda$):

$$
\begin{aligned}
\mathbb{E}[\chi_{\text{pos}}] &= \mathbb{E}\left[\frac{\sum_k \alpha_k^2 \hat{\lambda}_k}{\|\mathbf{y}\|^2 \text{Tr}(\mathbf{K})}\right] \approx \frac{\sum_k \mathbb{E}[\alpha_k^2]\lambda_k}{\mathbb{E}[\|\mathbf{y}\|^2]\sum_j \lambda_j} \\
&\propto \frac{\sum_k \lambda_k^2}{(\sum_k \lambda_k)^2}.
\end{aligned}
\tag{46}
$$

This result identifies $\chi_{\text{pos}}$ as the inverse *effective dimension* (or participation ratio) of the kernel spectrum.

### A.4.5. SPECTRAL TRANSFER AND FINAL SCALING

Finally, we relate the kernel eigenvalues $\lambda_k$ to the data covariance spectrum $\mu_k$. We assume the data spectrum follows a power law $\mu_k \sim k^{-\beta}$ with $\beta > 1$.

**Linear Dominance Principle.** The kernel $K$ admits a Mercer decomposition based on the Hermite expansion of the ReLU activation. The $n$-th order term in the expansion depends on the $n$-th tensor power of the data covariance. Crucially, for ReLU, the first Hermite coefficient is non-zero ($h_1 \neq 0$). Higher-order terms ($n \geq 2$) correspond to symmetric tensor powers of $\boldsymbol{\Sigma}$. For a data spectrum $\mu_k \propto k^{-\beta}$ with $\beta > 1$, the sorted eigenvalues of the $n$-th tensor power decay asymptotically faster than the linear term (Canatar et al., 2021; Scetbon & Harchaoui, 2021). Specifically, the spectral density of the product of power-law variables yields a tail behavior dominated by the linear component, establishing the linear dominance principle. Therefore, the kernel spectrum inherits the asymptotic decay rate of the data:

$$
\lambda_k \sim \mu_k \sim k^{-\beta}.
\tag{47}
$$

**Final Scaling Law.** Substituting $\lambda_k \propto k^{-\beta}$ into the expression for spectral position:

$$
\chi_{\text{pos}} \propto \frac{\sum_{k=1}^{\infty} k^{-2\beta}}{\left(\sum_{k=1}^{\infty} k^{-\beta}\right)^2}.
\tag{48}
$$

This concludes the derivation, explicitly linking the initial optimization dynamics to the spectral exponent $\beta$ via $\chi_{\text{pos}}$.

## B. Experimental Details

### B.1. Computation of LNP-Components

In this section, we provide further details on the computational aspects of measuring the LNP-components introduced in Section 3 and Appendix A.1.

**Efficient Computation without Explicit eNTK Formation.** We will now discuss how to compute the LNP-components efficiently without explicitly forming the empirical Neural Tangent Kernel (eNTK) matrix, which can be computationally prohibitive for large models. The spectral position component $\chi_{\text{pos}}^{AB}$ in the LNP-decomposition in Equation 24 naively requires the full eNTK computation, which would be prohibitively expensive for large-scale models. However, we can compute it indirectly by rearranging the LNP-decomposition:

$$\chi_{\text{pos}}^{AB} = \frac{\delta\mathcal{L}(A,B)}{\chi_{\text{loss}}^{AB} \cdot \chi_{\text{net}}^{AB}} \, . \tag{49}$$

This rearrangement allows us to efficiently obtain $\chi_{\text{pos}}^{AB}$ by computing the three quantities on the right-hand side separately.

We now describe how each of these components is measured in practice. The linearized loss change $\delta\mathcal{L}(A,B)$ is simply computed as the inner product of two parameter gradients

$$\delta\mathcal{L}(A,B) = \nabla_{\boldsymbol{\theta}}^T \mathcal{L}^B \, \nabla_{\boldsymbol{\theta}} \mathcal{L}^A \, , \tag{50}$$

which can be efficiently computed using standard backpropagation.

For the loss and network magnitudes $\chi_{\text{loss}}^{AB}$ and $\chi_{\text{net}}^{AB}$, we refer to their definition in Equation 24. Computing each of these components requires evaluating two terms: one on batch $A$ and one on batch $B$. For instance, the loss magnitude is given by $\chi_{\text{loss}}^{AB} = \left\|\nabla_f \mathcal{L}^B\right\|_2 \left\|\nabla_f \mathcal{L}^A\right\|_2$, requiring the computation of the loss gradient norm on both batches. Similarly, the network magnitude $\chi_{\text{net}}^{AB} = \left\|\nabla_\theta f^B\right\|_F \left\|\nabla_\theta f^A\right\|_F$ requires computing the Frobenius norm of the network Jacobian on both batches.

As shown in Equations 27 and 28, these magnitudes can be rewritten in terms of sample-wise gradient norms. In the field of differential privacy, efficient techniques have been developed to compute sample-wise gradient norms without explicitly forming the full Jacobian matrices $\nabla_\theta f^A$ and $\nabla_\theta f^B$ (Abadi et al., 2016; Goodfellow, 2015; Rochette et al., 2019; Lee & Kifer, 2020). Efficient implementations are available in popular deep learning frameworks such as Opacus (Yousefpour et al., 2022).

With this framework at hand, we can compute all three LNP-components efficiently without explicitly forming the eNTK matrix, enabling their application to large-scale models where eNTK computations become impractical.

**Hutchinson's Trace Estimation with Opacus (Yousefpour et al., 2022).** We implement the per-sample computation of the LNP-components using the Opacus library. While this allows for very efficient sample-wise gradient norms of one-dimensional functions, a naive implementation would require separate backward passes for each output dimension to calculate the sample-wise gradients $\nabla_{\boldsymbol{\theta}} f(\mathbf{x}_i)$ when the network outputs are multi-dimensional. This can significantly increase the computational cost when the output dimension $m$ is large, as e.g. in language modeling where it is equal to the vocabulary size i.e. $m > 10^4$. To avoid computing separate backward passes for each output dimension, we apply Hutchinson's trace estimator (Hutchinson, 1990) to estimate the squared norm of the sample-wise gradients with respect to multi-dimensional outputs. Mathematically, we estimate the squared norm of the sample-wise gradients as

$$\left\|\nabla_{\boldsymbol{\theta}} f^A(\mathbf{x}_i)\right\|_F^2 = \text{Tr}\left(\nabla_{\boldsymbol{\theta}} f^A(\mathbf{x}_i) \nabla_{\boldsymbol{\theta}}^T f^A(\mathbf{x}_i)\right) \approx \frac{1}{M}\sum_{m=1}^M v_m^T \nabla_{\boldsymbol{\theta}} f^A(\mathbf{x}_i) \nabla_{\boldsymbol{\theta}}^T f^A(\mathbf{x}_i)\, v_m = \frac{1}{M}\sum_{m=1}^M \left(v_m^T \nabla_{\boldsymbol{\theta}} f^A(\mathbf{x}_i)\right)^2 \, , \tag{51}$$

where $v_m \in \mathbb{R}^m$ are random Rademacher vectors (entries sampled from $\{-1,1\}$ with equal probability) and $M$ is the number of random vectors used for the estimation. This approach allows us to estimate the squared norm of the sample-wise gradients with only $M$ backward passes per batch, significantly reducing the computational overhead compared to computing exact sample-wise gradients for multi-dimensional outputs.

**Convergence of the Hutchinson Estimator.** We choose $M = 10$ Rademacher projections as a practical trade-off between accuracy and per-step cost; Figure 6 validates this choice. The single-step coefficient of variation of the estimated squared

gradient norm is approximately 23% at $M = 10$ (top right). Because the LNP-components reported in this work are averaged over windows of 50 training steps with independent Rademacher draws at each step, the effective coefficient of variation reduces to $23\%/\sqrt{50} \approx 3.3\%$, comparable to the 3.1% obtained at $M = 500$ projections per step but at $50\times$ lower per-step cost. The fitted $1/\sqrt{M}$ decay (top right) matches the theoretical Hutchinson rate. The convergence study was conducted on a trained 7.2M-parameter model using 32 training samples. The bottom panels confirm the practical consequence: $\chi_{\text{pos}}$ agrees tightly across 3 independent training seeds for all model sizes on both SimpleStories and CIFAR-5M.

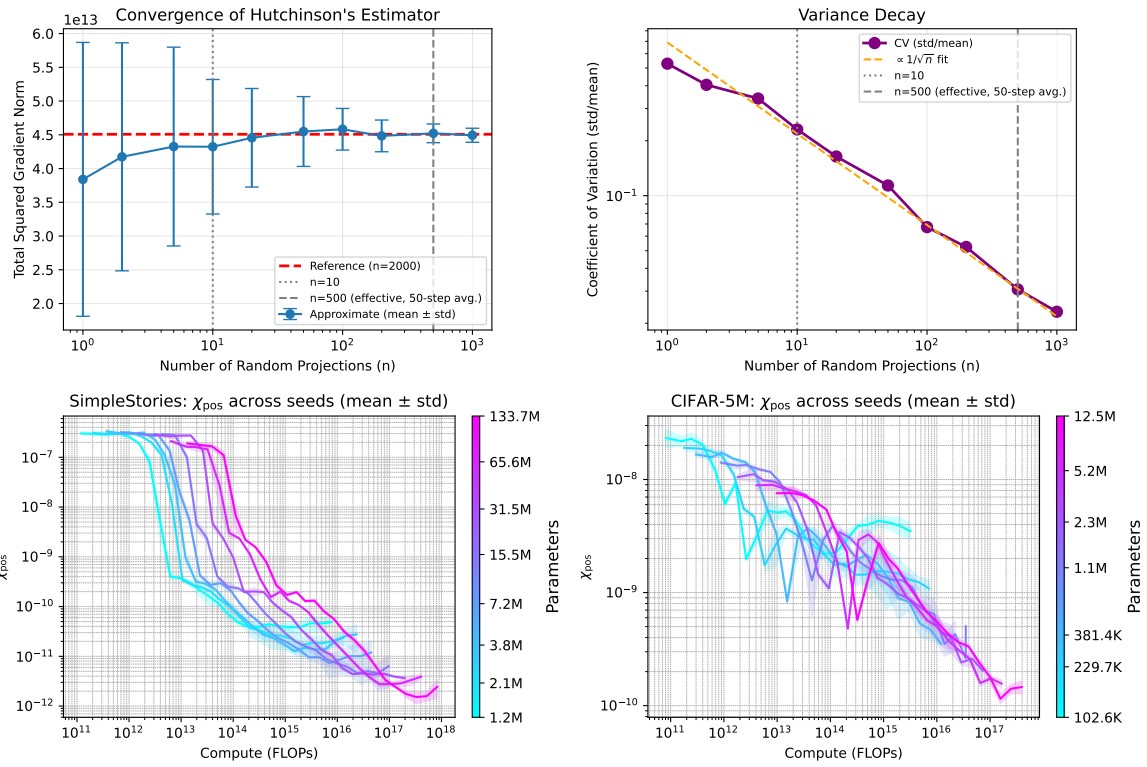

*Figure 6.* Convergence of the Hutchinson estimator for the squared gradient norm. **(Top left)** Estimated total squared gradient norm as a function of the number of projections $M$, with the converged reference at $M = 2000$ (dashed). Error bars show one standard deviation over 50 runs. **(Top right)** Coefficient of variation as a function of $M$ with fitted $1/\sqrt{M}$ decay, on a trained 7.2M-parameter model using 32 training samples. Vertical lines mark $M = 10$ (used throughout this work) and $M = 500$ (matching the effective accuracy after 50-step window averaging). **(Bottom)** Standard deviation of $\chi_{\text{pos}}$ across 3 training seeds vs. compute for all model sizes on SimpleStories (left) and CIFAR-5M (right). The tight cross-seed agreement at $M = 10$ confirms that this choice provides sufficient accuracy in practice.

**Estimated Computational Overhead.**    Using the techniques described above, we can compute the LNP-components with minimal computational overhead during training. In this section, we will show how to estimate the additional computational cost incurred by computing the LNP-components alongside standard training. First, we note that the LNP-components are not required to be computed at every training step. Instead, we compute them across windows of $W$ subsequent training steps. We choose these windows to be located at logarithmically spaced intervals during training, as the dynamics of training typically evolve more slowly at later stages. While we provide measurement details in Appendix B.2, we will use a typical setting to illustrate the computational overhead. We choose $W = 50$ training steps per window and $S = 20$ windows across training, resulting in a total of $S \cdot W = 1000$ training steps with LNP measurements.

Next, we analyze the computational cost of computing the LNP-components at each training step within a window. The main additional cost arises from computing the sample-wise gradient norms using Hutchinson's trace estimator with $M$ random vectors. This requires $M$ forward and backward passes per batch, and as explained in Appendix B.1, we need to compute these norms for both batches $A$ and $B$, resulting in a total of $2M$ forward and backward passes for computing the loss and network magnitudes. Additionally, we need to compute the inner product of the two parameter gradients to obtain $\delta\mathcal{L}(A, B)$, which requires two additional forward and backward passes. Overall, this results in a total of $2 + 2M$ forward and backward passes per training step with LNP measurements.

Assume that we train for a total of $T = 100000$ requiring the same number of forward and backward passes. If we choose $M = 10$ random vectors for the Hutchinson estimator, the total number of additional forward and backward passes for computing the LNP-components across all windows is

$$S \cdot W \cdot (2 + 2M) = 20 \cdot 50 \cdot (2 + 20) = 22000 \,. \tag{52}$$

Thus, the total number of forward and backward passes during training with LNP measurements rises from $100,000$ to $122,000$, resulting in an estimated computational overhead of only $22\%$.

**Memory Overhead.** Storing sample-wise gradients for large models is efficiently handled by Opacus using gradient checkpointing and memory-efficient backpropagation techniques (Yousefpour et al., 2022). In the current implementation, the main memory overhead arises from storing a second parameter gradient in memory. This occurs when computing the inner product of the two parameter gradients to obtain $\delta\mathcal{L}(A, B) = \nabla_{\theta}^T \mathcal{L}^B \nabla_{\theta} \mathcal{L}^A$. Consequently, we obtain a memory increase of approximately $1\times$ the model size.

**Comparison to Direct eNTK Methods.** The computational savings of the LNP-decomposition come from its structure, not from Hutchinson estimation alone. As shown in Section 3, $\chi_{\text{pos}}$ can be written as the scalar ratio $\delta\mathcal{L}/(\chi_{\text{loss}} \cdot \chi_{\text{net}})$, where $\delta\mathcal{L} = \|\nabla_{\theta}\mathcal{L}\|^2$ is a single scalar and $\chi_{\text{loss}}, \chi_{\text{net}}$ reduce to sums of *squared* per-sample gradient norms. The cross-sample structure of the eNTK enters only implicitly through $\delta\mathcal{L}$; no per-sample Jacobian and no kernel entry is ever materialized.

Methods that probe the eNTK directly do not admit this reduction. For example, kernel-target alignment (Cristianini et al., 2001; Baratin et al., 2021) requires the cross-sample entries of $\Theta$ explicitly, which in turn requires constructing per-sample Jacobians $J \in \mathbb{R}^{nm \times N}$ at $O(nm)$ backward passes per batch, where $n$ is the batch size, $m$ the per-sample output dimension (vocab size $\times$ sequence length for language models), and $N$ the number of model parameters. For LLMs with large vocab and long context, $nm$ reaches $10^8$ or beyond, making direct eNTK computation prohibitive in both compute and memory.

By contrast, our per-step cost is $O(M)$ forward-backward passes (with $M = 10$ in our experiments), *independent of $nm$*. For typical LLM settings, this amounts to a difference of several orders of magnitude per measured step.

## B.2. Setups

In this section, we provide further details on the experimental setups used in our study. We choose the SimpleStories (Finke et al., 2025) and CIFAR-5M (Nakkiran et al., 2020) datasets because their modest computational requirements allow extensive experiments across multiple model sizes and training runs.

### B.2.1. SIMPLESTORIES

The SimpleStories dataset is a collection of 2 million short stories with a custom tokenizer that reduces the vocabulary size to 4096 tokens (Finke et al., 2025). We train Llama 2 transformer models of varying sizes (1.2M, 2.1M, 3.8M, 7.2M, 15.5M, 31.5M, 65.6M, 134M parameters) on the next-token prediction task (Touvron et al., 2023). Model details are provided in Table 1. We train each model for a single epoch using AdamW with weight decay 0.01, $\beta_1 = 0.9$, and $\beta_2 = 0.95$. We use

*Table 1.* Model details for the Llama 2 transformer models trained on SimpleStories.

| Layers | Hidden Dim | FF Dim | Heads | Params |
|--------|-----------|--------|-------|--------|
| 4 | 96 | 256 | 2 | 1.2M |
| 4 | 128 | 512 | 2 | 2.1M |
| 5 | 196 | 768 | 3 | 3.8M |
| 6 | 256 | 768 | 4 | 7.2M |
| 7 | 384 | 1024 | 5 | 15.5M |
| 8 | 512 | 1536 | 8 | 31.5M |
| 18 | 512 | 1536 | 8 | 65.6M |
| 18 | 768 | 2048 | 12 | 134M |

learning rates (1e-2, 5e-3, 5e-3, 4e-3, 2e-3, 6e-4, 3e-4, 1.5e-4) for the different model sizes (in order of increasing size), selected via preliminary grid searches. We apply a linear learning-rate warmup over the first 100 steps. We use batch size 32,

context length 512 tokens, and train for a total of 66116 steps. Experiments are repeated for 3 random model initializations, preserving the data order across runs.

We measure the LNP-components across logarithmically spaced windows during training. Specifically, we use $S = 24$ windows, each spanning $W = 50$ training steps. Because we compute the LNP-components only once within overlapping windows, this results in a total of 801 training steps with LNP measurements. We use $M = 10$ random vectors for the Hutchinson estimator to compute sample-wise gradient norms efficiently.

To compute the LNP-components from Equation 24, we use the current training batch for dataset batch $A$. For dataset batch $B$, we use a fixed validation set of 160 samples, from which we draw a batch of 32 samples for each LNP computation within a window. Using the analysis from Appendix B.1, this setup results in an estimated computational overhead of approximately 27% during training.

For the linear probe experiments, we use (i) the final model checkpoints obtained after training on SimpleStories and (ii) randomly initialized models as backbones. We then freeze the backbone and train a linear classifier on top of the final-layer representations using SimpleStories. All other model and training (hyper-)parameters are kept the same as in the main training setup to ensure a fair comparison of the learned representations.

### B.2.2. CIFAR-5M NEXT PIXEL

The CIFAR-5M dataset is a collection of 6 million (5 M train and 1 M test) generated CIFAR-10-like images (Nakkiran et al., 2020). We convert the images into grayscale flattened sequences of 1024 pixels with values in $\{0, \ldots, 255\}$ and use a vocabulary size of 256 tokens for the next-pixel prediction task. We train Llama 2 transformer models of varying sizes (102.6K, 229.7K, 381.4K, 1.1M, 2.3M, 5.2M, 12.5M parameters) on the next-pixel prediction task (Touvron et al., 2023). Model details are provided in Table 2. We train each model for a single epoch using AdamW with weight decay 0.01,

*Table 2.* Model details for the Llama 2 transformer models trained on the CIFAR-5M next-pixel prediction task.

| Layers | Hidden Dim | FF Dim | Heads | Params |
|--------|-----------|--------|-------|--------|
| 3 | 32 | 256 | 2 | 102.6K |
| 3 | 64 | 256 | 2 | 229.7K |
| 3 | 96 | 256 | 2 | 381.4K |
| 4 | 128 | 512 | 2 | 1.1M |
| 5 | 192 | 512 | 3 | 2.3M |
| 6 | 256 | 768 | 4 | 5.2M |
| 7 | 384 | 1024 | 5 | 12.5M |

$\beta_1 = 0.9$, and $\beta_2 = 0.95$. We use learning rates (4e-2, 3e-2, 7e-3, 5e-3, 4e-3, 2e-3, 2e-3) for the different model sizes (in order of increasing size), selected via preliminary grid searches. We apply a linear learning-rate warmup over the first 100 steps. We use batch size 128, context length 1024 tokens, and train for a total of 42207 steps. Experiments are repeated for 3 random model initializations, preserving the data order across runs.

We measure the LNP-components across logarithmically spaced windows during training. Specifically, we use $S = 23$ windows, each spanning $W = 50$ training steps. Because we compute the LNP-components only once within overlapping windows, this results in a total of 752 training steps with LNP measurements. We use $M = 10$ random vectors for the Hutchinson estimator to compute sample-wise gradient norms efficiently.

To compute the LNP-components from Equation 24, we use the current training batch for dataset batch $A$. For dataset batch $B$, we use a fixed validation set of 4096 samples, from which we draw a batch of 128 samples for each LNP computation within a window. Using the analysis from Appendix B.1, this setup results in an estimated computational overhead of approximately 39% during training.

For the linear probe experiments, we use (i) the final model checkpoints obtained after training on CIFAR-5M and (ii) randomly initialized models as backbones. We then freeze the backbone and train a linear classifier on top of the final-layer representations using the CIFAR-5M next-pixel prediction task. All other model and training (hyper-)parameters are kept the same as in the main training setup to ensure a fair comparison of the learned representations.

### B.2.3. CIFAR-5M IMAGE CLASSIFICATION

The CIFAR-5M dataset is a collection of 6 million generated CIFAR-10-like images (Nakkiran et al., 2020). We use the RGB images with $32 \times 32$ pixels for an image classification task with 10 classes.

We train Vision Transformer (ViT) models of varying sizes (546.2K, 1.8M, 9.4M parameters) on the image classification task (Dosovitskiy et al., 2020). Model details are provided in Table 3.

*Table 3.* Model details for the Vision Transformer models trained on the CIFAR-5M image classification task.

| Patch Size | Hidden Dim | MLP Dim | Heads | Layers | Params |
|---|---|---|---|---|---|
| 4 | 128 | 256 | 4 | 4 | 546.2K |
| 4 | 192 | 768 | 4 | 4 | 1.8M |
| 4 | 360 | 1440 | 6 | 6 | 9.4M |

We train each model for 5 epochs using AdamW with weight decay 0.05, $\beta_1 = 0.9$, and $\beta_2 = 0.999$. We use learning rates (2.86e-4, 1.89e-4, 9.0e-5) for the different model sizes (in order of increasing size), selected via preliminary grid searches. We use batch size 128, image size $32 \times 32$ with patch size 4, and train for a total of 211034 steps.

We measure the LNP-components across logarithmically spaced windows during training. Specifically, we use $S = 51$ windows, each spanning $W = 50$ training steps. Because we compute the LNP-components only once within overlapping windows, this results in a total of 2550 training steps with LNP measurements. We do not use Hutchinson's trace estimator here, as the output dimension is small (10 classes) and we can compute sample-wise gradient norms exactly without significant overhead.

To compute the LNP-components from Equation 24, we use the current training batch for dataset batch $A$. For dataset batch $B$, we use a fixed validation set of 600268 samples, from which we draw a batch of 128 samples for each LNP computation within a window.

Using the analysis from Appendix B.1, this setup results in an estimated computational overhead of approximately 27% during training.

### B.2.4. CIFAR-5M IMAGE CLASSIFICATION UNDER MUP

To disentangle width from feature-learning intensity (see Appendix D.3 for the motivation), we replicate the CIFAR-5M classification experiments of Appendix B.2.3 under maximal-update parameterization (muP) (Yang et al., 2021; 2022). We use the same CIFAR-5M dataset as in the standard runs (Appendix B.2.3).

We train Vision Transformer models under muP at the sizes in Table 4. We apply muP following Yang et al. (2021; 2022) with

*Table 4.* Model details for the muP Vision Transformer models trained on the CIFAR-5M image classification task.

| Patch Size | Hidden Dim | MLP Dim | Heads | Layers | Params |
|---|---|---|---|---|---|
| 4 | 96 | 384 | 3 | 4 | 459.7K |
| 4 | 128 | 512 | 4 | 4 | 809.4K |
| 4 | 192 | 768 | 6 | 4 | 1.8M |
| 4 | 256 | 1024 | 8 | 4 | 3.2M |
| 4 | 384 | 1536 | 12 | 4 | 7.1M |

base width $d_{\text{base}} = 192$ and base learning rate $\eta_{\text{base}} = 1.57e-4$, tuned via a grid search at $d_{\text{base}}$; all other hyperparameters are identical to the standard-parameterization runs (Appendix B.2.3). This also applies to the LNP measurement setup.

### B.2.5. RANDOM FEATURE MODEL DETAILS

This setup is designed to validate the theoretical predictions in Equation 11 by creating a controlled environment where the LNP-components follow analytically tractable scaling behavior.

**Data Generation.** We generate synthetic Gaussian data $\mathbf{X} \in \mathbb{R}^{n \times d}$ with $n = 10000$ samples and input dimension $d = 128$. The data covariance matrix $\boldsymbol{\Sigma} = \text{diag}(\boldsymbol{\mu})$ is diagonal with eigenvalues following a power-law decay: $\mu_k \propto k^{-\beta}$ for $k = 1, \ldots, d$, where $\beta \in [1.0, 3.0]$ controls the spectral decay rate. We normalize eigenvalues such that $\sum_{k=1}^{d} \mu_k = d$. Each sample is generated as $\mathbf{x}_i = \mathbf{z}_i \odot \sqrt{\boldsymbol{\mu}}$, where $\mathbf{z}_i \sim \mathcal{N}(0, \mathbf{I}_d)$ and $\odot$ denotes element-wise multiplication, yielding $\mathbf{x}_i \sim \mathcal{N}(0, \boldsymbol{\Sigma})$.

**Student Architecture.** We employ a random feature model (RFM) with $m = 5000$ hidden units and frozen first-layer weights trained via squared loss:

$$f_{\mathbf{c}}(\mathbf{x}) = \frac{1}{\sqrt{m}} \sum_{j=1}^{m} c_j \cdot \sigma(\mathbf{w}_j^\top \mathbf{x}), \tag{53}$$

where $\mathbf{W} = [\mathbf{w}_1, \ldots, \mathbf{w}_m]^\top \in \mathbb{R}^{m \times d}$ is the frozen projection matrix with entries $W_{ji} \sim \mathcal{N}(0, d^{-1})$, $\sigma(\cdot) = \sqrt{2} \cdot \text{ReLU}(\cdot)$ is the variance-preserving ReLU activation, and $\mathbf{c} \in \mathbb{R}^m$ are trainable readout weights initialized at zero.

**RKHS-Realizable Teacher.** The teacher function is constructed to lie exactly within the student's reproducing kernel Hilbert space (RKHS):

$$f^*(\mathbf{x}) = \frac{1}{\sqrt{m}} \sum_{j=1}^{m} c_j^* \cdot \sigma(\mathbf{w}_j^\top \mathbf{x}), \tag{54}$$

where crucially, $\{\mathbf{w}_j\}_{j=1}^{m}$ are the *same* frozen projection vectors used by the student model, ensuring equal capacity and eliminating approximation errors that would arise from teacher-student capacity mismatch. The teacher coefficients are drawn as $\mathbf{c}^* \sim \mathcal{N}(0, \mathbf{I}_m)$.

This construction ensures *realizability*: the target function $f^*$ is exactly representable within the function class of the student. Both models share the same feature map $\boldsymbol{\phi}(\mathbf{x}) = \frac{1}{\sqrt{m}}[\sigma(\mathbf{w}_1^\top \mathbf{x}), \ldots, \sigma(\mathbf{w}_m^\top \mathbf{x})]^\top \in \mathbb{R}^m$, and thus inhabit the same RKHS with kernel $k(\mathbf{x}, \mathbf{x}') = \boldsymbol{\phi}(\mathbf{x})^\top \boldsymbol{\phi}(\mathbf{x}')$. With this normalization, $k(\mathbf{x}, \mathbf{x}') \to K(\mathbf{x}, \mathbf{x}')$ as $m \to \infty$, where $K$ is the population kernel of Appendix A.4.

**Evaluation Protocol.** Labels are generated as $\tilde{\mathbf{y}} = f^*(\mathbf{X})$ and normalized to unit variance: $\mathbf{y} = (\tilde{\mathbf{y}} - \bar{\tilde{\mathbf{y}}})/\text{std}(\tilde{\mathbf{y}})$. We evaluate the model at initialization (before any training) to isolate the effect of the data spectrum and kernel structure. We compute the LNP-components with respect to the squared loss $\mathcal{L}(\mathbf{c}) = \frac{1}{2n}\|\mathbf{y} - \boldsymbol{\Phi}\mathbf{c}\|_2^2$, where $\boldsymbol{\Phi} = [\boldsymbol{\phi}(\mathbf{x}_1), \ldots, \boldsymbol{\phi}(\mathbf{x}_n)]^\top \in \mathbb{R}^{n \times m}$ is the feature matrix with the zero-initialized readout layer. Each experiment is repeated across 100 random seeds (varying $\mathbf{W}$, $\mathbf{c}^*$, and $\mathbf{X}$) to estimate the expected scaling behavior of $\chi_{\text{net}}$, $\chi_{\text{loss}}$, and $\chi_{\text{pos}}$ as a function of $\beta$.

## C. Evaluation Details

### C.1. Data Processing and Visualization

**Loss Curve Smoothing.** The training loss values shown in Figure 1 are smoothed using an exponential moving average (EMA) with a smoothing factor of 0.1 for visualization purposes (McKinney, 2010). This reduces noise while preserving the overall trends in the loss dynamics.

**LNP-Component Visualization.** The LNP-components displayed in Figures 1 and 5 are computed over the measurement windows specified in Section B.2. For Figure 1, we plot the full dynamics without averaging for the first 40 training steps to capture the initial behavior. After this point, we plot the average value computed over windows of 50 steps to provide a clearer view of the longer-term trends.

**Handling Overfitting in Linear Probes.** For the linear probe experiments discussed in Section 5 and presented in Figure 4, we observe that models tend to overfit in later training stages. This overfitting leads to a reduction in the spectral position $\chi_{\text{pos}}$ that can obscure the natural reduction caused by learning smaller eigenvalue modes. To mitigate this effect, we apply the following strategy: from each measurement window, we select only the top 10% of spectral position values. Since this filtering is applied uniformly across all models, it does not introduce bias in the comparative analysis. Specifically,

Equation 12 is evaluated by computing the ratio

$$\frac{\chi_{\mathrm{pos}}(\mathrm{random})}{\chi_{\mathrm{pos}}(\mathrm{pre\text{-}trained})} = \frac{\langle \chi_{\mathrm{pos}}^{\mathrm{top}\ 10\%}(\mathrm{random}) \rangle_{\tau > 0.9t}}{\langle \chi_{\mathrm{pos}}^{\mathrm{top}\ 10\%}(\mathrm{pre\text{-}trained}) \rangle_{\tau > 0.9t}}, \tag{55}$$

where $\tau$ denotes the normalized training time (ranging from 0 to 1) and $\langle \cdot \rangle$ represents averaging over the selected data points within the specified time windows.

## D. Supplementary Experimental Results

### D.1. Test Loss Scaling

This section provides the test loss scaling curves corresponding to the training loss results shown in Figure 1 (top). Figure 7 demonstrates that the scaling laws observed for training loss extend to test performance: test loss follows similar power-law relationships with model size for both SimpleStories (left) and CIFAR-5M next-pixel prediction (right). The power-law fits show comparable exponents and $R^2$ values to those observed for training loss, confirming that larger models achieve better generalization performance and that the scaling behavior is not limited to fitting the training data.

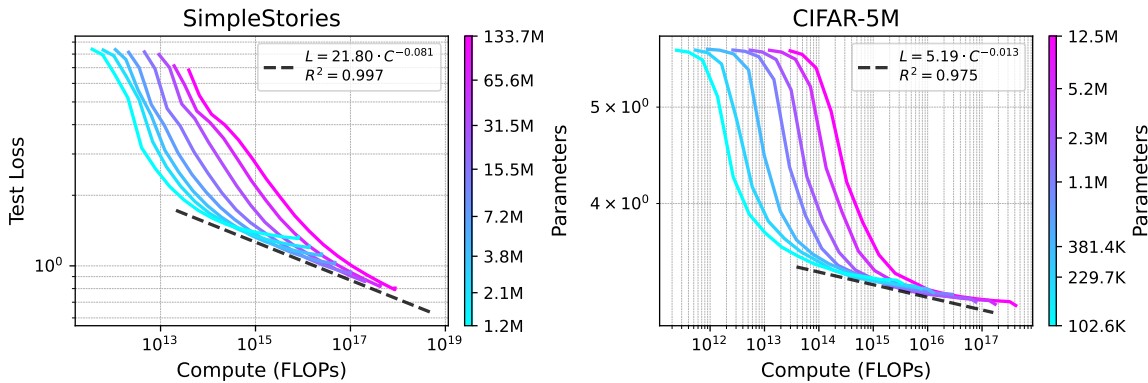

*Figure 7.* Test loss scaling laws for Llama language models on SimpleStories (left) and next-pixel prediction on CIFAR-5M (right). Test loss follows power-law relations with model size, similar to the training loss shown in Figure 1. The dashed lines indicate power-law fits with exponents and $R^2$ values. Larger models achieve lower test losses, confirming that scaling behavior extends to generalization performance.

### D.2. Compute-Optimal Data/Model Trade-Off

The main paper presents scaling experiments along the model-size axis under a fixed single-epoch training recipe (Appendices B.2.1 and B.2.2). Here we examine how spectral position responds to a different slicing: at a fixed total compute budget, how does the choice between a larger model with less data and a smaller model with more data affect test loss and $\chi_{\mathrm{pos}}$?

Figure 8 traces these isocompute curves: test loss (left) and spectral position (right) as a function of model size at fixed FLOP budgets, on SimpleStories (top) and CIFAR-5M (bottom). Each color-coded curve corresponds to a single compute budget; points along a curve correspond to models of different sizes trained with that budget.

The qualitative behavior of $\chi_{\mathrm{pos}}$ tracks that of test loss across both datasets: along each isocompute curve, $\chi_{\mathrm{pos}}$ moves with the loss as the data/model ratio shifts. The two axes play complementary roles in spectral descent: more data provides more gradient steps for $\chi_{\mathrm{pos}}$ to migrate into the spectral tail, while larger models lower the floor that $\chi_{\mathrm{pos}}$ can ultimately reach (Section 4). At a fixed compute budget, $\chi_{\mathrm{pos}}$ and loss therefore respond to the data/model trade-off in tandem: both reflect the same balance between training time and capacity.

### D.3. Vision Transformer Classification: Standard and muP Parameterization

Beyond the language and next-pixel-prediction settings of the main text, we evaluate spectral reach on CIFAR-5M image classification with Vision Transformers (Appendix B.2.3). The standard-parameterization panel of Figure 9 (left) shows

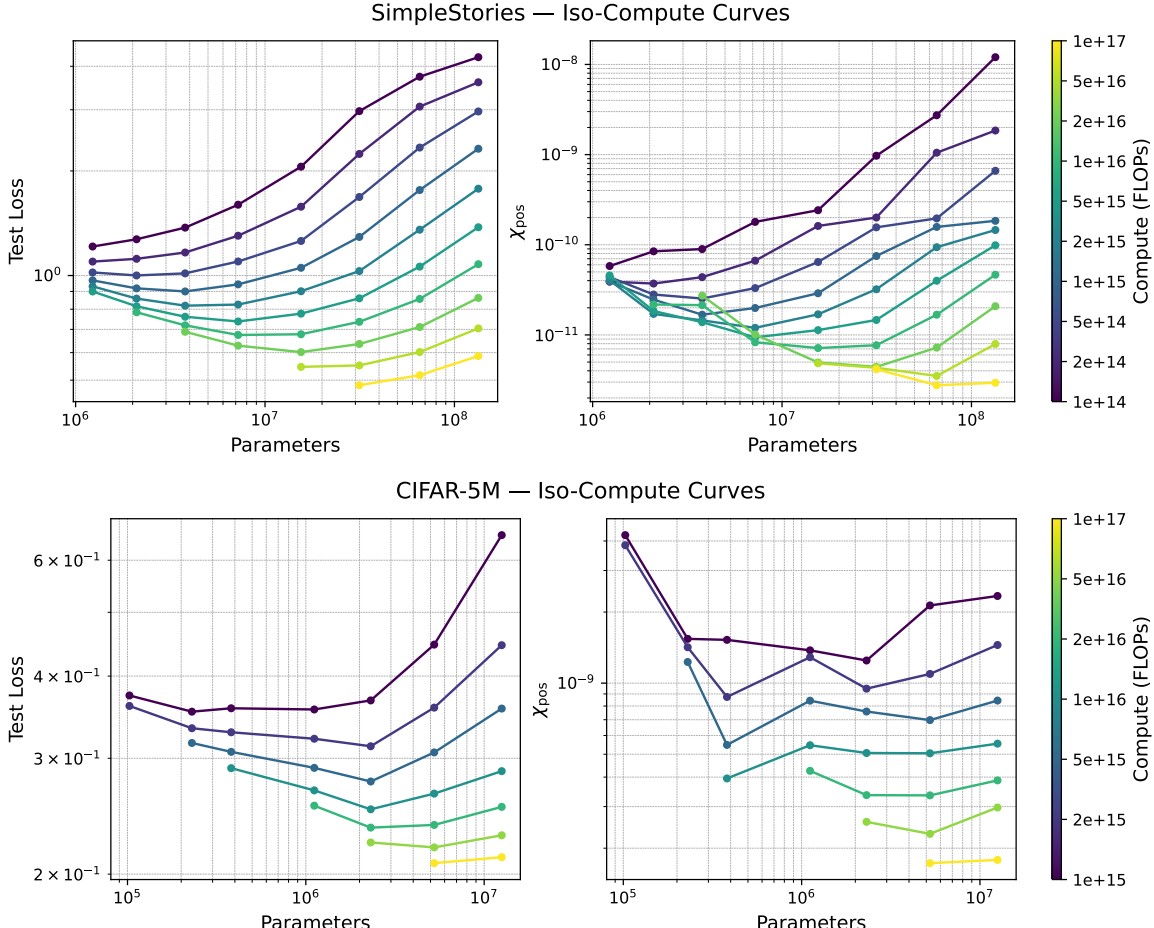

*Figure 8.* Isocompute curves on SimpleStories (top) and CIFAR-5M (bottom). Test loss (left) and spectral position $\chi_{\text{pos}}$ (right) as a function of model size at fixed compute budgets (color-coded FLOP totals). Each curve traces models of different sizes trained with the same total compute. Spectral position qualitatively follows the same isocompute trend as test loss across both datasets, suggesting that spectral reach scales predictably with the compute-optimal data/model ratio.

that the spectral-position phenomenon generalizes to vision classification: training loss decreases with model size, spectral position decreases throughout training, and larger ViTs reach lower $\chi_{\text{pos}}$ values.

Under standard parameterization, however, width and feature-learning intensity are entangled: changing model width also changes the rate at which representations evolve during training. The scale ordering observed in the main text could in principle be driven by either factor. muP (Yang et al., 2021; 2022) resolves this entanglement by rescaling weight initializations, learning rates, and (where applicable) attention temperatures so that feature-learning intensity is held constant across widths. If the ordering of spectral reach across model sizes persists under muP, it must be attributable to model capacity rather than width-dependent feature-learning intensity.

The muP panel of Figure 9 (right) shows that the ordering is indeed preserved: larger ViTs continue to reach lower $\chi_{\text{pos}}$ values. We conclude that the scale ordering reported in Section 4 is capacity-driven, not intensity-driven. Setup details for the muP runs are provided in Appendix B.2.4.

### D.4. Dynamics of all LNP-Components

The main paper focuses on the dynamics of spectral position $\chi_{\text{pos}}$ (Section 4) and the gradient-magnitude scale $\chi_{\text{net}}$ (Section 5). Here we report the dynamics of all three LNP-components, $\chi_{\text{loss}}$, $\chi_{\text{net}}$, and $\chi_{\text{pos}}$, as well as the product $\chi_{\text{net}} \cdot \chi_{\text{pos}}$, for all model sizes on both SimpleStories (Figure 10) and CIFAR-5M next-pixel prediction (Figure 11).

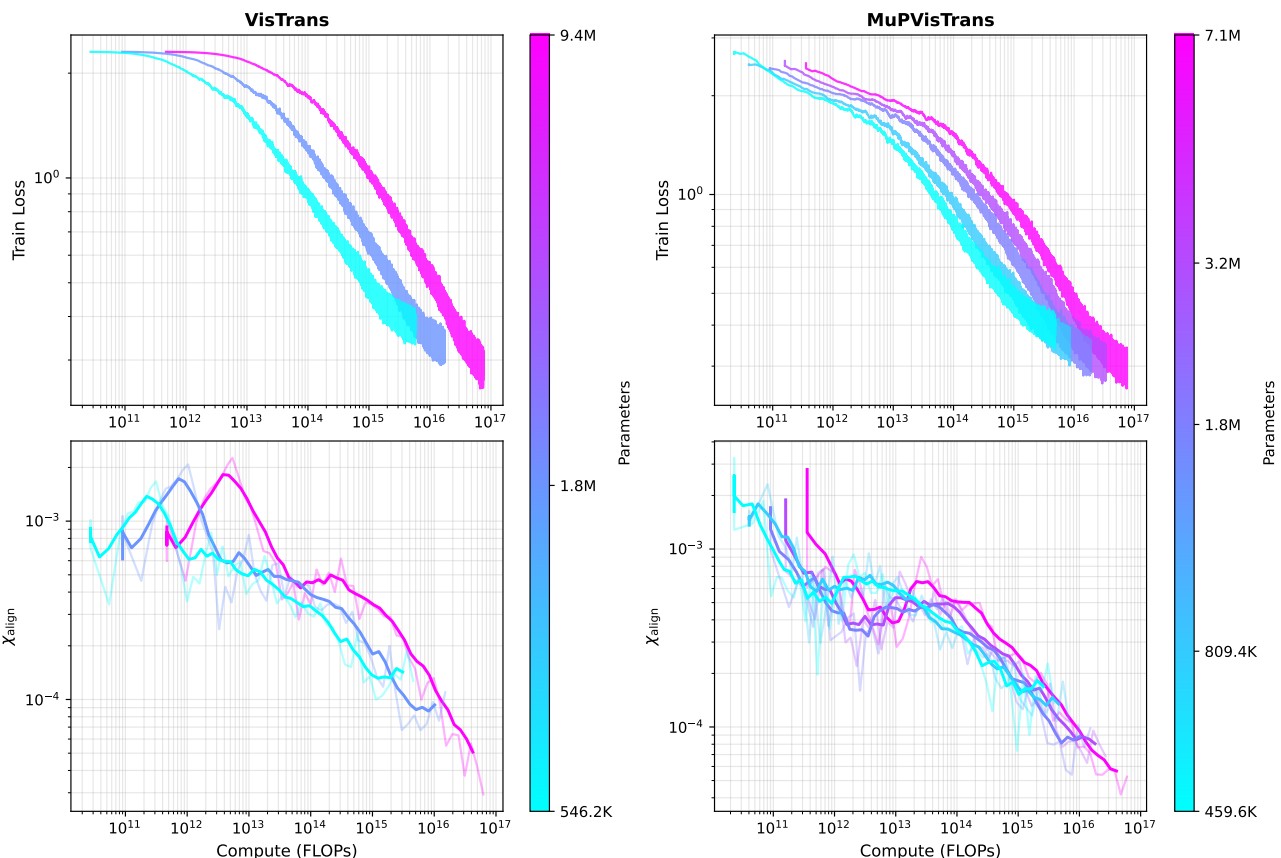

*Figure 9.* Side-by-side comparison of training loss and spectral position $\chi_{\text{pos}}$ vs. compute for ViT models trained on CIFAR-5M under standard parameterization (left) and muP (right). The left panel shows that the spectral-reach phenomenon extends to image classification: larger ViTs reach lower $\chi_{\text{pos}}$ values throughout training, mirroring the behavior observed in the language and next-pixel-prediction settings. The muP panel shows that this ordering is preserved when feature-learning intensity is held constant across widths, indicating that the scaling is attributable to model capacity rather than feature-learning intensity. We approximate compute as FLOPs $\approx 6 \times$ Params $\times$ Processed Patches. An exponential moving average with smoothing factor 0.05 is applied to the loss curves.

**Loss-magnitude dynamics.** By definition, $\chi_{\text{loss}} = \|\nabla_f \mathcal{L}\|_2^2$ depends only on the loss function and the current prediction errors, so it should mirror the loss curve up to model-independent factors. The top-left panels of both figures confirm this: $\chi_{\text{loss}}$ tracks the loss closely across all model sizes throughout training (compare to the top panels of Figure 1).

**Spectral position decrease versus network-magnitude increase.** Because $\chi_{\text{net}} = \text{Tr}(\Theta)$ grows during training in feature-learning models, one might worry that the observed decrease in $\chi_{\text{pos}}$ is a passive consequence of trace inflation: as the kernel trace increases, the normalized spectral position would have to compensate. The bottom-right panels of Figures 10 and 11 rule out this explanation. The product $\chi_{\text{net}} \cdot \chi_{\text{pos}}$ more closely resembles $\chi_{\text{pos}}$ (bottom left) than $\chi_{\text{net}}$ (top right), indicating that the $\chi_{\text{pos}}$ decrease substantially exceeds the $\chi_{\text{net}}$ increase. A compensation between the two would produce an approximately constant product, which the data rule out. This is consistent with the direction-of-causality argument in Section 6, in which spectral bias drives the $\chi_{\text{pos}}$ decrease and the $\chi_{\text{net}}$ increase is its secondary, complementary response.

### D.5. Explicit eNTK Eigendecomposition

The decomposition $\chi_{\text{pos}} = \sum_k \gamma_k^2 p_k$ (Section 3) implies that a decrease in $\chi_{\text{pos}}$ over training corresponds, mathematically, to a shift of the projection coefficients $\gamma_k^2$ toward smaller normalized eigenvalues $p_k$. Here we verify this shift empirically by directly computing $\gamma_k^2$ and $p_k$ from an explicit eNTK eigendecomposition.

To make the eNTK tractable, we evaluate it on a reduced output space: a single SimpleStories sample with $K = 40$ vocabulary indices and $S = 256$ token positions, on the 3.8M-parameter Llama model from the main experiments. We

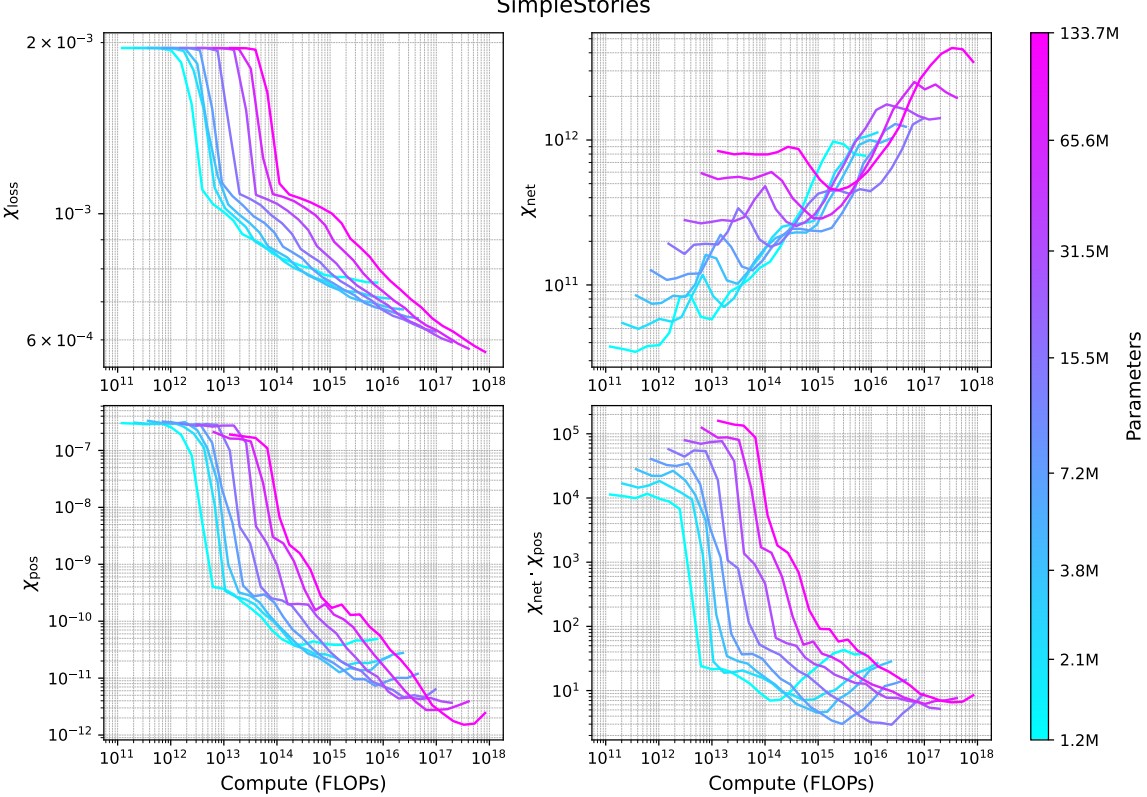

*Figure 10.* Dynamics of all LNP-components for Llama models trained on SimpleStories. Shown are $\chi_{\text{loss}}$ (top left), $\chi_{\text{net}}$ (top right), $\chi_{\text{pos}}$ (bottom left), and their product $\chi_{\text{net}} \cdot \chi_{\text{pos}}$ (bottom right) vs. compute for all model sizes. $\chi_{\text{loss}}$ mirrors the loss dynamics; the product $\chi_{\text{net}} \cdot \chi_{\text{pos}}$ is dominated by the $\chi_{\text{pos}}$ decrease.

compare the spectra at initialization and after training.

Figure 12 reports the eigenvalue distribution $p_k$, the squared projection coefficients $\gamma_k^2$, and their product $\gamma_k^2 p_k$ at the two checkpoints. After training, $\gamma_k^2$ is softly reweighted toward smaller eigenvalues, consistent with the mathematical prediction. The effect is moderate because of the constrained setup: the eNTK here is computed over roughly $6000\times$ fewer tokens than in the main experiments, which compresses the spectral statistics toward the bulk of the distribution and reduces the separation between dominant and subordinate modes. We expect the shift to be substantially more pronounced at the scales used in the main analysis, where the full eNTK eigendecomposition is computationally infeasible.

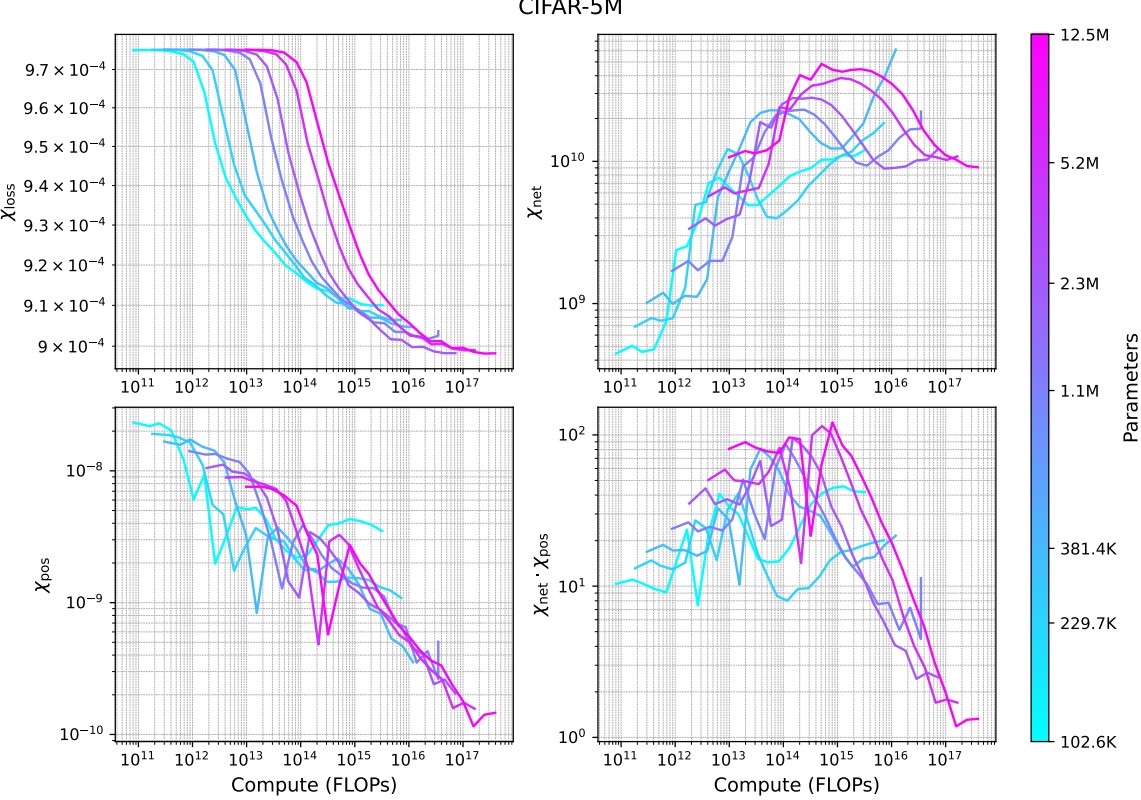

*Figure 11.* Dynamics of all LNP-components for Llama models trained on CIFAR-5M next-pixel prediction. Shown are $\chi_{\text{loss}}$ (top left), $\chi_{\text{net}}$ (top right), $\chi_{\text{pos}}$ (bottom left), and their product $\chi_{\text{net}} \cdot \chi_{\text{pos}}$ (bottom right) vs. compute for all model sizes. $\chi_{\text{loss}}$ mirrors the loss dynamics; the product $\chi_{\text{net}} \cdot \chi_{\text{pos}}$ is dominated by the $\chi_{\text{pos}}$ decrease.

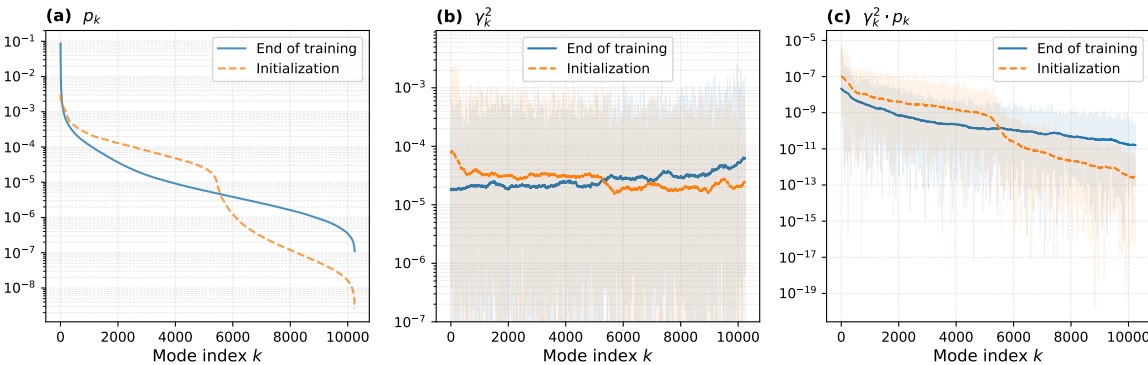

*Figure 12.* Explicit eNTK spectral distributions for a 3.8M-parameter Llama model on SimpleStories at initialization (dashed) and after training (solid). **(a)** Eigenvalue distribution $p_k$. **(b)** Squared projection coefficients $\gamma_k^2$. **(c)** Product $\gamma_k^2 p_k$. The eNTK is computed in a reduced output space ($K = 40$ vocabulary indices, $S = 256$ token positions) on a single sample due to the cost of full materialization. After training, $\gamma_k^2$ is softly reweighted toward smaller eigenvalues, consistent with the $\chi_{\text{pos}}$ decrease reported in the main paper.

