# OpenReview forum: "Spectral Reach: Understanding Neural Scaling as Progress into the Spectral Tail"
_ICML.cc/2026/Conference — ICML 2026 regular_

### Official Review · Reviewer_Sz77 · 2026-03-12

**Soundness:** 2
**Presentation:** 4
**Significance:** 1
**Originality:** 4
**Overall Recommendation:** 3
**Confidence:** 4

**Summary:**

- A metric is proposed, the alignment between the eNTK and loss residuals, as a tool to understand the learning dynamics of deep nets.
- This quantity is deemed intractable, thus an efficient estimator is proposed, that involves decomposing the alignment between 3 terms, LNA: loss, network, alignment
- The estimator is validated on an analytical model where xhi_loss and xhi_net are essentially constants
- An empirical investigation of the training dynamics of Llama models shows how the quantities evolves throughout training, and the effect of increasing the number of parameters (scaling laws). In particular, it is shown that the loss alignment is lower for models with more parameters.

**Compliance With Llm Reviewing Policy:**

Affirmed.

**Key Questions For Authors:**

1. On the relevance of the loss alignment metric

The metric misses a side-by-side comparison with NTK alignment as studied in previou works: what new insight does it bring ? In which case is it more relevant to measure loss alignment rather than NTK alignment ?

2. On the relevance of the practical LNA estimator

"corresponding measurements are difficult to obtain at the scales where scaling laws are observed." l112 right please be more specific. I dont see any reason why eq.5, or NTK alignment, would be difficult to scale. Please also provide computational complexities for your proposed estimator.

3. Accuracy of the practical LNA estimator

I would have appreciated a comparison (at least empirical) with loss alignment (not the estimator) computed exactly on actual models (small ones) to get a sense of whether the non-linear correction is actually negligible, and how it evolves as training progresses ?

4. LNA estimator break down, trace(NTK)

How do all 3 xhi terms evolve simultaneously during training, and in particular:
is the decreasing loss aligment just a mere consequence of the trace(NTK) increase during training ?

"feature learning reshapes spectral magnitudes (χnet) to maintain learning progress as kernel alignment decreases" l83
"χnet can potentially increase to compensate, keeping the product δL sufficient to drive learning." l364 right
IMO this is the other way around: χnet increases and as a consequence, the quantity that you measure (loss alignment) decreases

5. Name of the metric

It would be misleading to call this quantity "kernel alignment" as it is different from what was previously called kernel alignment in kernel methods (Cortes and co) and NTK alignement papers.

6. And a minor comment, that won't change my evaluation: there seems to be a pattern in loss alignment (Figure 1 bottom) with a sudden drop (more pronounced in CIFAR-5M). Do you maybe have an intuition regarding that ?

**Limitations:**

yes

**Strengths And Weaknesses:**

## Presentation and originality

are fine as far as I can tell.

## Soundness and significance

I can't make up my mind about the relevance of the proposed metric. What new insight can you obtain that you could not without this metric. How exactly does it differ from previous measured NTK aligment with the task studied in other (properly referenced) papers ? It looks to me that the decrease in loss alignment observed is only a consequence of an increasing trace(NTK) as you observe in figure 5 (and was previously observed e.g. in Baratin et al. 2021). Is it really intractable to compute the quantity exactly rather than using the proposed estimator ?

---

> ### Author Rebuttal · Authors · 2026-03-30
>
> *Additional figures are provided in the [supplementary material (SM)](https://anonymous.4open.science/r/ICML26_rebuttle_plots-01A8/main.pdf).*
>
> > Q1: On the relevance of the loss alignment metric
>
> We thank the reviewer for the chance to clarify this. The key difference is what each metric measures. Previous NTK alignment metrics (e.g., Baratin et al.) measure how much the kernel has aligned with *fixed*, pre-defined task-relevant directions. They answer “has the model adapted to the task?” Our $\chi_{\text{align}}$ instead measures alignment with the *current* loss gradient, which changes over training. It answers a different question: “where in the eNTK spectrum is learning currently focused?” (see also our response to Reviewer fgpN, W2 on naming).
>
> Our experiments show that $\chi_{\text{align}}$ systematically decreases with model scale, meaning larger models access deeper into the eNTK spectral tail — and achieve lower loss. This motivates a concrete reframing: "how to reduce loss" becomes "how to make the spectral tail accessible," suggesting interventions (e.g., spectral reshaping, targeted gradient scaling) that previous alignment metrics cannot motivate (see also Reviewer fgpN, Q3).
>
> > Q2: Scalability of LNA computation vs. NTK alignment. Computational complexities.
>
> We should have been more precise. Evaluating Eq. 5 exactly, as well as computing NTK alignment (e.g., Baratin et al.), requires per-sample Jacobians $J \in \mathbb{R}^{nc \times p}$, where $n$ is the batch size, $c$ the output dimension (vocab size $\times$ sequence length), and $p$ the number of parameters. Even without materializing the full eNTK ($\Theta = JJ^T \in \mathbb{R}^{nc \times nc}$), obtaining the Jacobian costs $O(nc)$ backward passes per batch. For LLMs with large vocab and sequence length, $nc$ is on the order of $10^8+$ (and explicit $J$ or $\Theta$ construction is also memory-prohibitive).
>
> Our computation bypasses the Jacobian entirely. We only require per-sample gradient *norms*, obtained via Hutchinson's trace estimator with $M$ random projections, each costing one forward-backward pass. Per measured step, this amounts to $2 + 2M$ forward-backward passes (with $M=10$: 22 passes). The key scaling advantage is that this cost is independent of $nc$, whereas NTK-based methods scale as $O(nc)$ per step. For typical LLM settings, this is a difference of several orders of magnitude. We are happy to include this complexity comparison in the revised manuscript for clarification.
>
> > Q3: Accuracy of the LNA computation.
>
> We want to clarify that the $\chi$ terms are not approximations. They are instantaneous properties of the network at each point in training, computed exactly via Eq. 5. The nonlinear term $\nu$ does not enter their definition or computation; it only determines how well the linear prediction of the next loss value tracks the actual loss change. Computing $\chi$ correctly does not require $\nu$ to be small. The only estimation involved is the Hutchinson trace estimator, whose accuracy we validate in a convergence study (see our response to Reviewer fgpN, Q1 and Fig. 1 in SM).
>
> > Q4: Is decreasing $\chi_{\text{align}}$ just a consequence of $\text{Tr}(\text{NTK})$ increasing?
>
> In Figures 5 and 6 (see SM), we now show all three $\chi$ terms over training, as well as the product $\chi_{\text{net}} \cdot \chi_{\text{align}}$. The data shows that the decrease in $\chi_{\text{align}}$ is substantially stronger than the increase in $\chi_{\text{net}}$. If $\chi_{\text{net}}$ growth were driving the alignment decrease, these effects would cancel. Prior work on spectral bias establishes that models learn from large to small eNTK eigenvalues. Since $\chi_{\text{align}}$ measures the expected eigenvalue currently being learned, its decrease is a direct consequence of spectral bias rather than a secondary effect. We believe the increase in $\chi_{\text{net}}$ is the secondary response, driven by adapting representations (see also Reviewer fgpN, Q3). We note that stronger compensation could in principle occur in other architectures/initializations, which we plan to investigate in future work. We are happy to include Figure 5 and 6 in the revised appendix.
> > Q5: Name of the metric
>
> We agree. See our response to Reviewer fgpN (W2), where we discuss the connection to kernel alignment and rename the quantity to $\chi_{\text{pos}}$ (*spectral position*) in the revised paper.
>
> > Q6: Pattern in loss alignment (sudden drop).
>
> Thanks for pointing this out. We believe this corresponds to a restructuring of the eNTK: while spectral bias predicts a consistent decrease in $\chi_{\text{align}}$, feature learning can reorganize the eNTK eigenvalue distribution as new data is processed. The sudden drop likely reflects such a restructuring event, where the model adapts its representations in a way that shifts the spectral distribution.

---

> > ### Author Rebuttal · Reviewer_Sz77 · 2026-04-07
> >
> > Q1: I am still not so convinced. If the interventions you are referring to arise from the new analysis provided by the xhi_align metric, then perhaps these should be actually implemented.
> >
> > Q2: the Hutchinson estimator thus looks to be the main reason why you claim your metric to be less computationally demanding. I don't see any particular reason why it would apply differently to other quantities (NTK alignment)
> >
> > Q3: Ok
> >
> > Q4: I don't see the figures that you refer to.
> >
> > Q5: Ok
> >
> > Overall, I think the paper would be strenghtened by more comprehensive answers to these reviews. I am keeping my score

---

> > > ### Author Response · Authors · 2026-04-07
> > >
> > > We thank the reviewer for the follow-up.
> > >
> > > **Q2 (computational cost).** The scaling advantage does not come from Hutchinson itself, but from the structure of the LNA decomposition. As shown in Eq. 10 of the manuscript, $\chi_{\text{pos}}$ can be expressed as the ratio $\delta\mathcal{L} / (\chi_{\text{loss}} \cdot \chi_{\text{net}})$, where $\delta\mathcal{L}$ is the *scalar* linearized loss change, and $\chi_{\text{loss}}, \chi_{\text{net}}$ reduce to sums of per-sample gradient *norms*. The latter are obtainable via ghost-norm techniques (Goodfellow, 2015) at the cost of a few extra backward passes, with no per-sample Jacobian or kernel entry ever materialized. The cross-sample structure of the eNTK is instead captured implicitly through $\delta\mathcal{L}$.
> > >
> > > By contrast, prior NTK alignment work (e.g., Cristianini et al., 2001; Baratin et al., 2021) constructs the kernel (or large submatrices) explicitly: the kernel-target alignment $\langle \Theta, yy^T\rangle / (\|\Theta\|_F \|yy^T\|_F)$ does not admit a reduction to per-sample norm quantities, and the cross-sample entries of $\Theta$ must be probed directly.
> > >
> > > The practical consequence is that our per-step cost is $O(M)$ forward-backward passes ($M$ being the number of Hutchinson projections, $M=10$ in our experiments), *independent* of the output dimension $nc$ (batch size $\times$ sequence length $\times$ vocabulary). Prior NTK alignment formulations instead scale with $nc$, which for typical LLM settings ($nc \sim 10^8$) amounts to a difference of several orders of magnitude per measured step.
> > >
> > > **Q4 (figures).** Figures 5 and 6 are in the supplementary PDF, linked at the top of our initial response.
> > >
> > > **Q1 (interventions).** We want to clarify the distinction between the contribution of this paper and the directions it opens up. The central finding is empirical and, we believe, substantive on its own: $\chi_{\text{pos}}$ exposes a previously unmeasured, scale-dependent property of LLM training. Larger models systematically reach deeper into the eNTK spectral tail, and this depth tracks loss across scales. Establishing this phenomenon required developing the LNA decomposition, validating its tractability at LLM scale, and demonstrating the scaling behavior across model sizes and datasets. None of this depends on any intervention.
> > >
> > > The interventions we sketched in the initial response are *implications* of this finding, not claims we make in the paper. Each (e.g., spectral reshaping via initialization or parameterization) is a substantial direction with its own methodological choices, and we believe it deserves a dedicated treatment rather than a single bundled experiment. We are actively pursuing this as follow-up work, and will revise the manuscript to clearly separate the established result from the directions it motivates.
> > >
> > > We hope these clarifications are useful for the record.

---

### Official Review · Reviewer_fgpN · 2026-03-13

**Soundness:** 2
**Presentation:** 3
**Significance:** 2
**Originality:** 3
**Overall Recommendation:** 5
**Confidence:** 3

**Summary:**

This paper studies the evolution of the alignment of the empirical neural tangent kernel eNTK throughout training on tasks that generate commonly observed powerlaw neural scaling curves. They introduce a decomposition for the (linearized) change in the loss $\delta \mathcal L$ that contains a contribution from the loss gradient, the trace of the NTK, and an alignment metric $\chi_{\text{align}}$. They show that in the lazy training regime on powerlaw random features, the $\chi_{\text{align}} = \frac{ \text{Tr} \Theta^2}{(\text{Tr} \Theta)^2}$ measures an inverse participation ratio for the NTK features which is constant in the lazy training regime. They then track their alignment measures as a function of training in the non-lazy regime. In this case, they find that the $\chi_{\text{align}}$ *decreases* over the course of training. They argue that this enhances the spectral reach of the model, the ability of the model to learn low-lying eigenmodes of the kernel. As if to compensate for the decreasing alignment, the kernel trace often increases throughout training. They find that the change in alignment from a random baseline decreases with model size.

**Compliance With Llm Reviewing Policy:**

Affirmed.

**Final Justification:**

The authors have now tested against confounds related to parameterization with experiments varying width in muP. I think the paper provides a potentially useful tool (beyond even the present paper) to track how feature kernels evolve throughout training. While the theory is not causally complete, it may be providing diagnostics for feature learning that are computable in large networks. I am thus now in favor of acceptance.

**Key Questions For Authors:**

1. The Hutchison estimator uses a very small number of random vectors $M=10$. Have the authors verified, either experimentally or numerically, that this leads to accurate results and bounded variance of the estimation?

2. Have the authors experimented with large learning rates where the linearization could break down? Do models exhibit edge of stability or are those effects suppressed due to minibatch SGD?

2. Sometimes the authors write as if $\chi_{\text{align}}$ causally impacts the learning dynamics on lower level eigenmodes. The order of causality is not completely clear to me. Could it not be the case that the gradient dynamics and current loss level on modes being learned late in training cause a decrease in $\chi_{\text{align}}$? Are there ways to causally intervene (maybe gradient scaling or learning rate schedules) to manipulate $\chi_{net}$, $\chi_{\text{loss}}$ etc in order to measure the impact on the spectral reach effect?

**Limitations:**

The authors acknowledge the limitations from linearization and studying two datasets. They could also potentially acknowledge the causality claims (see question 3 above).

**Strengths And Weaknesses:**

***Strengths***

**Novel decomposition**
The decomposition of the linearized loss in to $\chi_{loss}, \chi_{net}, \chi_{align}$ is quite novel to the best of my knowledge. It can be computed on a variety of architectures and datasets and there is a clear interpretation of each term in the lazy limit.

**Computationally Efficient Metric**
Because $\delta L$, the eNTK trace $\chi_{net}$, and $\chi_{loss}$ can be computed efficiently, this method avoids explicit construction of the eNTK.

***Weaknesses***

**Entangled Width and Feature Learning Effects**

A central comparative claim is that larger models exhibit greater spectral reach, and separately that feature learning enhances spectral reach. However, in standard parameterization (NTK or SP), the degree of feature learning itself changes with width in a non-trivial way. In the NTK parameterization, feature learning vanishes in the infinite-width limit; in standard parameterization, the feature learning strength scales with width in a manner that is neither controlled nor uniform across model sizes.

Without adopting maximal update parameterization (μP, Yang et al., 2021) or equivalently a mean-field parameterization that holds feature learning strength constant as width scales, it is impossible to cleanly attribute the observed $\chi_{\text{align}}$ differences across model sizes to capacity versus feature learning intensity (larger models happen to be in a more feature-learning-rich regime under SP). The two effects are fully entangled in the current experimental design. A clean ablation would train models of varying width under $\mu$P, where the feature learning strength is held constant across scales, and examine whether the $\chi_{\text{align}}$ ordering persists. If it does not, the paper's interpretation of spectral reach as a capacity effect could be suspect (such as in Figure 4).

**Interpretation of $chi_{\text{align}}$ as an Alignment metric**

The $\chi_{\text{align}}$ does not correspond to a cosine similarity or classic kernel aligment between two matrices which would require normalization in the following sense $\text{sim}(A,B) = \frac{ \text{Tr} A B}{ \sqrt{ \text{Tr} A^2 \text{Tr} B^2 } }$. More importantly, in the mini-batch setting $\chi_{\text{align}}$ can be negative, breaking the [0,1] bound entirely and making the geometric interpretation (projection onto eigenmodes) ill-defined. The paper would benefit from either (a) clearly repositioning $\chi_{\text{align}}$ as a spectral moment rather than an alignment, or (b) providing a proper normalized alignment metric and treating the current quantity as a computationally efficient proxy.

**Questions about Validity of Loss Linearization**

The decomposition focuses on the linearized loss. It would be potentially useful to show that the nonlinear effects are not relevant in the experimental regimes of interest (namely that step sizes are not large enough for higher order terms to matter). There are potentially issues if models are operating close to the edge of stability (close to maximal learning rates).

---

> ### Author Rebuttal · Authors · 2026-03-30
>
> *Additional figures are provided in the [supplementary material (SM)](https://anonymous.4open.science/r/ICML26_rebuttle_plots-01A8/main.pdf).*
>
> > Weakness 1: Entangled Width and Feature Learning Effects
>
> The reviewer is correct that, under standard parameterization, width and feature-learning intensity are entangled, and we thank them for encouraging us to investigate this further. We have conducted additional experiments under muP (Yang et al., 2021), which holds feature learning strength constant across widths. In Figure 2 (see SM), we directly compare standard and muP parameterizations and show that the spectral reach ordering across model sizes is preserved under muP. This confirms that the effect is attributable to model capacity rather than width-dependent feature learning intensity. We will include these results in the revised paper.
>
> > Weakness 2: Interpretation of $\chi_{\text{align}}$ as an Alignment metric
>
> $\chi_{\text{align}}$ is closely related to the kernel alignment of Cristianini et al. (2001): $\chi_{\text{align}} = \mathrm{Sim}(\nabla_f \mathcal{L} (\nabla_f \mathcal{L})^T, \Theta) \cdot \lVert\Theta\rVert_F / \mathrm{Tr}(\Theta)$, differing only by a spectral concentration factor of the eNTK. We agree, however, that the name "alignment" invites expectations that this quantity does not fully meet. In the revised paper, we therefore rename it $\chi_{\text{pos}}$ (*spectral position*), the expected normalized eNTK eigenvalue under loss-gradient weighting, indicating where in the spectrum learning is currently focused. For the sake of continuity with the original submission, we use $\chi_{\text{align}}$ in this response.
>
> > Weakness 3: Questions about Validity of Loss Linearization
>
> We want to clarify an important distinction: the LNA components are defined via spatial derivatives at the current parameter values. They are instantaneous properties of the network at a given point in parameter space, independent of step size or the amount of non-linearity. The decomposition is an exact mathematical identity, so computing the $\chi$ terms does not require the dynamics to be linear or the step size to be small.
>
> > Q1: Verification of Hutch with $M=10$.
>
> We conducted a convergence study (Fig. 1, see SM): At $n = 10$, the per-step CV is approximately 23%, but since we average over windows of 50 training steps (each with independent Rademacher draws), the effective CV is reduced to $\approx 3.1$% at $n = 500$ projections per step, but at $50\times$ lower per-step cost. We chose $n = 10$ as a practical trade-off between accuracy and cost, and validate this choice empirically: the bottom panels show that $\chi_{\mathrm{align}}$ agrees tightly across 3 independent seeds for all model sizes on both datasets. We will add this analysis to the appendix.
>
>
> > Q2: Breakdown of linearizaton and EOS.
>
> We have not varied learning rates in this work, since as noted in our response to W3, the $\chi$ terms remain well-defined regardless of learning rate. The focus of this work is to analyze scaling behavior of language models under standard training conditions, where our measures provide a novel lens.
> We agree that the framework is well suited to study how learning rate affects the interplay between linear and nonlinear contributions to learning, as the LNA decomposition cleanly separates the two. Investigating how different learning rate regimes (including EoS) influence spectral reach is a direction we are excited to explore in future work, but it is beyond the scope of this paper. We will clarify this point in the revised manuscript.
>
> > Q3: Causality of $\chi_{\text{align}}$ decrease and interventions.
>
> **Causality.** Prior work on spectral bias establishes that models learn from large to small eNTK eigenvalues. Since $\chi_{\text{align}}$ (Equation 9) measures the expected eigenvalue currently being learned, its decrease is a direct consequence of spectral bias, not an independent effect. The increase in $\chi_{\text{net}}$ that compensates is therefore the secondary response, driven by adapting representations. We will clarify this in the manuscript.
>
> **Interventions.** This is an exciting direction that we also identify in our outlook (Section 6). If spectral reach is indeed what drives generalization, then the question "how do we improve generalization?" can be reframed as "how do we make the spectral tail accessible?" We see two complementary intervention strategies:
>
> 1. **Accelerate spectral descent:** use targeted interventions (gradient scaling, lr schedules, $\chi_{\text{net}}$ boosting) to reach subordinate modes faster.
> 2. **Reshape the spectrum:** compress the spectral tail into the bulk via architectural or initialization choices, so subordinate modes are easier to reach.
>
> We have not yet conducted experiments along either direction, but we believe this framing opens up a promising research direction. We are happy to expand the outlook in the revised paper to make these two strategies explicit.

---

> > ### Author Rebuttal · Reviewer_fgpN · 2026-04-04
> >
> > I appreciate the authors detailed response. I am still concerned about causality so I will make my score 4.

---

> > > ### Author Response · Authors · 2026-04-06
> > >
> > > We thank the reviewer for the continued engagement. We want to address the causality concern directly with the evidence at hand.
> > >
> > > We agree that we have not established causality through controlled intervention experiments. However, we believe the causal direction, that spectral bias drives the decrease in $\chi_{\text{align}}$, rather than the reverse, is strongly supported by three converging lines of evidence:
> > >
> > > 1. **Theoretical grounding.** Spectral bias (Rahaman et al., 2019; Cao et al., 2019) and subsequent analyses of kernel-regime learning dynamics (Bordelon et al., 2020; Basri et al., 2020) establish that neural networks learn from large to small eigenvalues. Since $\chi_{\text{align}}$ measures the expected eigenvalue under loss-gradient weighting, its decrease is a direct prediction of this theory, not a post-hoc interpretation.
> > >
> > > 2. **Ruling out confounds.** Our muP experiments (Fig. 2, SM) show that the spectral reach ordering across model sizes is preserved when feature-learning intensity is held constant, ruling out the most natural alternative explanation.
> > >
> > > 3. **Empirical decomposition.** In Figures 5 and 6 (SM), the decrease in $\chi_{\text{align}}$ substantially exceeds the increase in $\chi_{\text{net}}$. If the alignment decrease were a passive side effect of $\chi_{\text{net}}$ growth, these would cancel.
> > >
> > > We acknowledge that definitive causal claims would require intervention experiments (e.g., artificially clamping $\chi_{\text{align}}$ and observing effects on loss), which are beyond the scope of this work. We will revise the manuscript to clearly distinguish between the causal *direction* (supported by theory and converging evidence) and causal *mechanism* (which would require interventions to establish definitively).
> > >
> > > We respectfully note that the concerns raised in the initial review have each been addressed with clarification or new evidence (W1: muP experiments; W2: renaming of the metric; W3: mathematical identity; Q3: converging evidence from theory, controlled experiments, and empirical decomposition), and we hope the reviewer will consider whether the contribution of the paper merits reconsideration of the suggested score.

---

### Official Review · Reviewer_oqvA · 2026-03-13

**Soundness:** 4
**Presentation:** 4
**Significance:** 3
**Originality:** 3
**Overall Recommendation:** 4
**Confidence:** 3

**Summary:**

The paper proposes a new framework to measure the alignment between the eNTK and loss residuals with a low computational cost. Through a comprehensive analysis ranging from theoretical settings such as random feature networks to more realistic feature-learning regimes, the work demonstrates an unalignment behavior over training time that is consistent with spectral bias theory. The framework also explains why larger models tend to achieve lower training error through the concept of spectral reach, which describes the capacity to access eigenmodes with small eigenvalues of the eNTK. The experiments further confirm the spectral reach phenomenon in feature learning.

**Compliance With Llm Reviewing Policy:**

Affirmed.

**Final Justification:**

After the rebuttal and considering the other reviewers’ opinions, I will maintain my positive evaluation with a score of 4. The rebuttal addresses most of my concerns, but it is not strong enough for me to increase the score to 5.

**Key Questions For Authors:**

- The finding that dominant modes are learned first, and that once they are learned there is no further alignment in those directions, is interesting. Can this be empirically verified?

- What is the training dynamic of the term $\chi_{loss}$ over flops (similar as Figure 5)?

I'm happy to increase the score if these are discussed or addressed.

**Limitations:**

Yes

**Strengths And Weaknesses:**

**Strengths:**
- The presentation is excellent, the paper is logically structured and well motivated. The writing is clear and provides a high level of clarity.
- The proposed decomposition is simple, but elegant and has low computational overhead. The claims are well supported, either through a good theoretical framework or empirical feature learning experiments.
- The paper contains many interesting observations that provide deeper understanding of what feature learning is targeting and why scaling helps toward that goal.

**Weaknesses**:
- As discussed in the paper, the main concern is that the non-linear correction term is not analyzed and based on the middle panel in Figure 2, this term is quite significant.
- The data dimension is not discussed in the analysis. For instance, under a fixed computation budget and model size, it would be interesting to understand how increasing the dataset size affects the alignment behavior. Does larger dataset lead to higher kernel alignment?

---

> ### Author Rebuttal · Authors · 2026-03-30
>
> *Note: following feedback from other reviewers, we rename $\chi_{\text{align}}$ to $\chi_{\text{pos}}$ (spectral position) in the revised paper. We use $\chi_{\text{align}}$ throughout this response for consistency with the original submission. Additional figures are provided in the [supplementary material (SM)](https://anonymous.4open.science/r/ICML26_rebuttle_plots-01A8/main.pdf).*
>
> > W1: Significance of non-linear terms.
>
> We thank the reviewer for raising this point. We want to clarify that the LNA components are defined via spatial derivatives at the current parameter values. They are instantaneous properties of the network at a given point in parameter space. The decomposition is an exact mathematical identity, and the $\chi$ terms can be computed correctly regardless of how much nonlinearity is present in the dynamics.
>
> The nonlinear term $\nu$ captures what happens *after* a parameter update. While $\nu$ can potentially be significant, this does not affect the validity of the $\chi$ measurements. We acknowledge that the sketch in Figure 2 may draw undue attention to the nonlinear correction; the aim of this work is not to characterize $\nu$, but to show that the linear component alone, via the proposed $\chi$ measures, already provides actionable insights into scaling behavior. A full characterization of $\nu$ is an interesting direction for future work.
>
> > W2: Discussion of data dimension.
>
> We thank the reviewer for addressing this point. In standard LLM pretraining, models are trained for a single epoch without data repetition. In this setting, dataset size and training duration are equivalent: a "larger dataset" simply means more tokens processed. Our analysis in Section 4 already captures this effect: as training progresses (i.e., as more data is consumed), alignment decreases (see Figure 1 in the manuscript).
>
> The reviewer's question becomes distinct from training duration when multi-epoch or data-resampling strategies are used, where the same number of tokens can come from datasets of different sizes. While our current experiments do not isolate this variable, we can address the closely related question of how alignment behaves under compute-optimal data/model trade-offs. In Figure 3 (see SM), we present isocompute curves that show alignment alongside training loss as a function of model size at fixed compute budgets. Although we lack sufficient model sizes to fit a proper scaling law, alignment qualitatively follows the same isocompute trend as loss: The alignment reduces as more data comes in. It further suggests that it scales predictably with the compute-optimal data/model ratio. We will include this analysis in the revised paper.
>
>
> > Q1: Mode-wise alignment shift.
>
> This claim follows directly from the mathematical structure of the alignment. As shown in the paper, $\chi_{\text{align}} = \sum_k \gamma_k^2 \, p_k$, where $\gamma_k$ captures how the loss gradient projects onto the $k$-th eNTK eigenmode and $p_k$ is the normalized eigenvalue. A decreasing $\chi_{\text{align}}$ over training mathematically implies that the loss gradient's weight shifts from high-eigenvalue (dominant) to low-eigenvalue (subordinate) modes. This is what the quantity measures by definition.
>
> To provide additional empirical evidence, we performed an explicit eNTK eigendecomposition on a 3.8M-parameter Llama model trained on the Simplestories dataset. Due to the computational cost of materializing the full eNTK, we use a reduced output space (single sample, $K=40$ vocab indices, $S=256$ token positions). In Figure 4 (see SM), we compare the spectral distribution of $\gamma_k$ between a randomly initialized model and the same model after training. The results show a soft reweighting of $\gamma_k$ toward smaller eigenvalues after training, meaning the alignment is reduced for larger eigenvalues and increased for smaller eigenvalues. The effect is moderate due to the constrained setup; over 6000$\times$ fewer tokens than our main experiments. With limited data the spectral statistics are compressed toward the bulk of the distribution, reducing the separation between dominant and subordinate modes. We therefore expect the spectral shift to be substantially more pronounced at the scales used in our main analysis, where the full eNTK eigendecomposition is unfortunately not computationally feasible.
>
>
> > Q2: $\chi_{\text{loss}}$ over flops
>
> We agree this is a valuable addition. The dynamics of $\chi_{\text{loss}}$ over training were computed in our experiments but omitted from the manuscript due to space constraints. In Figures 5 and 6 (see SM), we show $\chi_{\text{loss}}$ alongside the other $\chi$ terms (also requested by Reviewer Sz77, Q4) and will include these plots in the revised paper. As expected from its definition, $\chi_{\text{loss}}$ closely resembles the behavior of the loss curve across all model sizes.

---

> > ### Author Rebuttal · Reviewer_oqvA · 2026-04-01
> >
> > I thank the authors for the detailed rebuttal and additional experiments. My concerns are mostly resolved. To clarify Q1, I meant the following: suppose we select two modes, one corresponding to a leading eigenvalue and one to a smaller eigenvalue. I would like to see the evolution of $\gamma_{k}^{2}$ over training time to verify this observation that dominant modes are learned first, and that once they are learned there is no further alignment in those directions.

---

> > > ### Author Response · Authors · 2026-04-06
> > >
> > > We thank the reviewer for the response and the clarification.
> > >
> > > The suggested direction is very interesting! Tracking $\gamma_k^2$ for selected eigenmodes over training would directly verify whether dominant modes are learned first and subsequently saturate in alignment.
> > > The central quantity we compute cheaply is $\chi_{\text{align}} = \sum_k \gamma_k^2 p_k$, which aggregates over all modes. Resolving individual $\gamma_k^2$ trajectories, however, requires the full eNTK eigendecomposition. The eNTK is a data-by-data matrix with dimensionality depending on batch size, sequence length, and vocabulary. For our training setup (batch size 32, sequence length 1024, vocabulary 512), this gives a $1,638,400 \times 1,638,400$ matrix, making eigendecomposition a real challenge in both memory and compute regardless of model size.
> > > The key methodological challenge is therefore not just computational cost, but the principled selection of individual modes: identifying which eigenmodes correspond to early- versus late-learned features without access to the full decomposition. This is precisely the problem we are currently developing methods to address. We will include a discussion of this direction in the revised paper.
> > >
> > > Given that the concerns raised in the initial review have been addressed (W1 with clarification of the LNA validity, W2 with isocompute analysis, Q1 with an explicit eigendecomposition, and Q2 with dynamics plots), we kindly ask the reviewer to consider whether the current score still reflects the state of the work after the rebuttal.

---

### Decision · Program_Chairs · 2026-04-30

**Decision:**

Accept (regular)

**Comment:**

This paper introduces the Loss-Network-Alignment (LNA) decomposition, factoring the instantaneous three interpretable scalars: a loss magnitude ($\chi_{\text{loss}}$), the eNTK trace ($\chi_{\text{net}}$), and a measurement of where eNTK spectrum the loss gradient currently concentrates ($\chi_{\text{align}}$, renamed $\chi_{\text{pos}}$ / spectral position in revision). A key methodological contribution is that this last quantity can be computed via per-sample gradient norms with a Hutchinson trace estimator, remaining tractable at LLM scale. Applied to Llama models on SimpleStories and to CIFAR-5M, the framework reveals a consistent pattern: larger and better-performing models sustain systematically lower spectral-position values throughout training, while $\chi_{\text{net}}$ grows to compensate. A linear-probing ablation demonstrates that this extended ``spectral reach'' is enabled by feature learning rather than width per se. The reviewers found the decomposition novel and potentially useful well beyond this paper, and the balance of opinion is in favor of acceptance.

*Requested revisions*

The authors should incorporate into the camera-ready the additions promised during the rebuttal. In particular:

1. The $\mu$P experiments comparing standard parameterization and maximal-update parameterization across model sizes, which disentangle model capacity from width-dependent feature-learning intensity and confirm that the spectral-reach ordering is preserved when feature-learning strength is held constant.

2. The renaming of $\chi_{\text{align}}$ to $\chi_{\text{pos}}$ / spectral position with a clear discussion that this quantity is a spectral moment, including an explicit acknowledgement that it can become negative.

3. A clearer treatment of the causal direction between decreasing spectral position and compensating $\chi_{\text{net}}$ growth, incorporating the evidence the authors marshalled (spectral bias, the $\mu$P control, and the asymmetric magnitudes of the $\chi_{\text{align}}$ decrease vs. $\chi_{\text{net}}$ increase).